# IMPROVED NOISE SCHEDULE FOR DIFFUSION TRAINING

## ABSTRACT

Diffusion models have emerged as the de facto choice for generating high-quality visual signals across various domains. However, training a single model to predict noise across various levels poses significant challenges, necessitating numerous iterations and incurring significant computational costs. Various approaches, such as loss weighting strategy design and architectural refinements, have been introduced to expedite convergence and improve model performance. In this study, we propose a novel approach to design the noise schedule for enhancing the training of diffusion models. Our key insight is that the importance sampling of the logarithm of the Signal-to-Noise ratio ($\log \text{SNR}$), theoretically equivalent to a modified noise schedule, is particularly beneficial for training efficiency when increasing the sample frequency around $\log \text{SNR} = 0$. This strategic sampling allows the model to focus on the critical transition point between signal dominance and noise dominance, potentially leading to more robust and accurate predictions. We empirically demonstrate the superiority of our noise schedule over the standard cosine schedule. Furthermore, we highlight the advantages of our noise schedule design on the ImageNet benchmark, showing that the designed schedule consistently benefits different prediction targets. Our findings contribute to the ongoing efforts to optimize diffusion models, potentially paving the way for more efficient and effective training paradigms in the field of generative AI.

## 1 INTRODUCTION

Diffusion models have emerged as a pivotal technique for generating high-quality visual signals across diverse domains, including image synthesis (Ramesh et al., 2022; Saharia et al., 2022; Rombach et al., 2022) , video generation (Ho et al., 2022; Singer et al., 2023; Brooks et al., 2024), and even 3D object generation (Wang et al., 2022; Nichol et al., 2022). One of the key strengths of diffusion models lies in their ability to approximate complex distributions, where Generative Adversarial Networks (GANs) may encounter difficulties. Despite the substantial computational resources and numerous training iterations required for convergence, improving the training efficiency of diffusion models is essential for their application in large-scale scenarios, such as high-resolution image synthesis and long video generation.

Recent efforts to enhance diffusion model training efficiency have primarily focused on two directions. The first approach centers on architectural improvements. For instance, the use of Adaptive Layer Normalization (Gu et al., 2022), when combined with zero initialization in the Transformer architecture Peebles & Xie (2023), has shown promising results. MM-DiT (Esser et al., 2024) extends this approach to multi-modality by employing separate weights for vision and text processing. Similarly, U-shaped skip connections within Transformers (Hoogeboom et al., 2023; Bao et al., 2022; Crowson et al., 2024) and reengineered layer designs (Karras et al., 2024) have contributed to more efficient learning processes.

The second direction explores various loss weighting strategies to accelerate training convergence. Works such as eDiff-I (Balaji et al., 2022) and Ernie-ViLG 2.0 (Feng et al., 2022) address training difficulties across noise intensities using a Mixture of Experts approach. Other studies have investigated prioritizing specific noise levels (Choi et al., 2022) and reducing weights of noisy tasks (Hang et al., 2023) to enhance learning effectiveness. Recent developments include a softer weighting

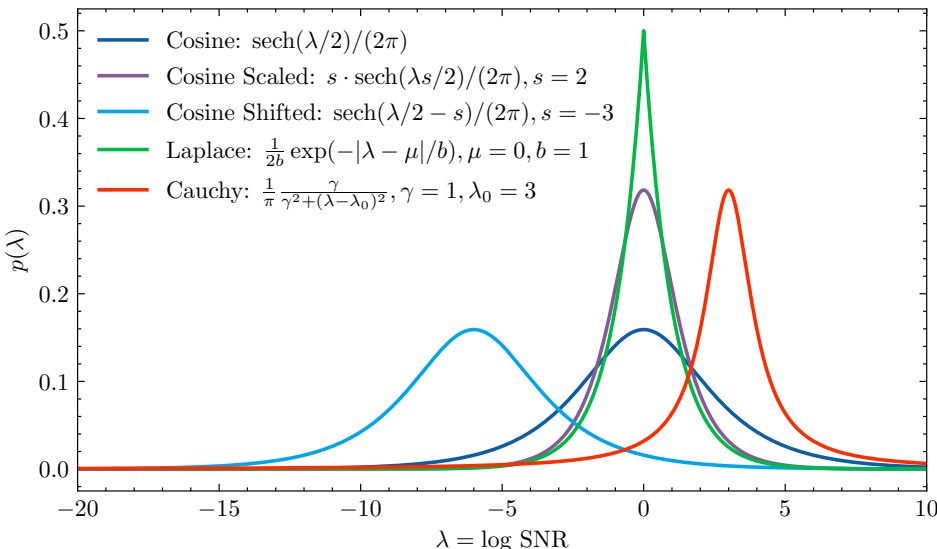

Figure 1: Illustration of the probability density functions of different noise schedules.

approach for high-resolution image synthesis (Crowson et al., 2024) and empirical findings on the importance of intermediate noise intensities (Esser et al., 2024).

Despite these advances, the fundamental role of noise scheduling in diffusion model training remains underexplored. In this study, we present a novel approach focusing on the fundamental role of noise scheduling, which is a function that determines how much noise is added to the input data at each timestep $t$ during the training process, controlling the distribution of noise levels that the neural network learns to remove. Our framework provides a unified perspective for analyzing noise schedules and importance sampling, leading to a straightforward method for designing noise schedules through the identification of curves in the $p(\lambda)$ distribution, as visualized in Figure 1. Through empirical analysis, we discover that allocating more computation costs (FLOPs) to mid-range noise levels (around $\log \mathrm{SNR} = 0$) yields superior performance compared to increasing loss weights during the same period, particularly under constrained computational budgets.

We evaluate several different noise schedules, including Laplace, Cauchy, and the Cosine Shifted/Scaled variants, through comprehensive experiments using the ImageNet benchmark with a consistent training budget of 500K iterations (about 100 epochs). Our results, measured using the Fréchet Inception Distance (FID) metric at both $256 \times 256$ and $512 \times 512$ resolutions, demonstrate that noise schedules with concentrated probability density around $\log \mathrm{SNR} = 0$ consistently outperform alternatives, with the Laplace schedule showing particularly favorable performance.

The key contributions of our work can be summarized as follows:

- A unified framework for analyzing and designing noise schedules in diffusion models, offering a more systematic approach to noise schedule optimization.
- Empirical evidence demonstrating the superiority of mid-range noise level focus over loss weight adjustments for improving training efficiency.
- Comprehensive evaluation and comparison of various noise schedules, providing practical guidelines for future research and applications in diffusion model training.

## 2 METHOD

### 2.1 PRELIMINARIES

Diffusion models (Ho et al., 2020; Yang et al., 2021) learn to generate data by iteratively reversing the diffusion process. We denote the distribution of data points as $\mathbf{x} \sim p_{\mathrm{data}}(\mathbf{x})$. The diffusion

process systematically introduces noise to the data in a progressive manner. In a continuous setting, the noisy data at timestep $t$ is defined as follows:

$$\mathbf{x}_t = \alpha_t \mathbf{x} + \sigma_t \boldsymbol{\epsilon}, \quad \text{where} \quad \boldsymbol{\epsilon} \sim \mathcal{N}(0, \mathbf{I}), \tag{1}$$

where $\alpha_t$ and $\sigma_t$ are the coefficients of the adding noise process, essentially representing the noise schedule. For the commonly used prediction target velocity: $\mathbf{v}_t = \alpha_t \boldsymbol{\epsilon} - \sigma_t \mathbf{x}$ (Salimans & Ho, 2022), the diffusion model $\mathbf{v}_\theta$ is trained through the Mean Squared Error (MSE) loss:

$$\mathcal{L}(\theta) = \mathbb{E}_{\mathbf{x} \sim p_{\text{data}}(\mathbf{x})} \mathbb{E}_{t \sim p(t)} \left[ w(t) \| \mathbf{v}_\theta(\alpha_t \mathbf{x} + \sigma_t \boldsymbol{\epsilon}, t, \mathbf{c}) - \mathbf{v}_t \|_2^2 \right], \tag{2}$$

where $w(t)$ is the loss weight, $\mathbf{c}$ denotes the condition information. In the context of class-conditional generation tasks, $\mathbf{c}$ represents the class label. Common practices sample $t$ from the uniform distribution $\mathcal{U}[0, 1]$. Kingma et al. (2021) introduced the Signal-to-Noise ratio as $\text{SNR}(t) = \frac{\alpha_t^2}{\sigma_t^2}$ to measure the noise level of different states. Notably, $\text{SNR}(t)$ monotonically decreases with increasing $t$. Some works represent the loss weight from the perspective of SNR (Salimans & Ho, 2022; Hang et al., 2023; Crowson et al., 2024). To simplify, we denote $\lambda = \log \text{SNR}$ to indicate the noise intensities. In the Variance Preserving (VP) setting, the coefficients in Equation 1 can be calculated by $\alpha_t^2 = \frac{\exp(\lambda)}{\exp(\lambda)+1}$, $\sigma_t^2 = \frac{1}{\exp(\lambda)+1}$.

While these foundational concepts have enabled significant progress in diffusion models, the choice of noise schedule remains somewhat ad hoc. This motivates us to develop a more systematic framework for analyzing and designing noise schedules by examining them from a probability perspective.

## 2.2 NOISE SCHEDULE DESIGN FROM A PROBABILITY PERSPECTIVE

The training process of diffusion models involves sampling timesteps $t$ from a uniform distribution. However, this uniform sampling in time actually implies a non-uniform sampling of noise intensities. We can formalize this relationship through the lens of importance sampling (Bishop & Nasrabadi, 2006). Specifically, when $t$ follows a uniform distribution, the sampling probability of noise intensity $\lambda$ is given by:

$$p(\lambda) = p(t) \left| \frac{\mathrm{d}t}{\mathrm{d}\lambda} \right| = -\frac{\mathrm{d}t}{\mathrm{d}\lambda}, \tag{3}$$

where the negative sign appears because $\lambda$ monotonically decreases with $t$. We take cosine noise schedule (Nichol & Dhariwal, 2021) as an example, where $\alpha_t = \cos\left(\frac{\pi t}{2}\right)$, $\sigma_t = \sin\left(\frac{\pi t}{2}\right)$. Then we can deduce that $\lambda = -2\log\tan(\pi t/2)$ and $t = 2/\pi \arctan e^{-\lambda/2}$. Thus the distribution of $\lambda$ is: $p(\lambda) = -\mathrm{d}t/\mathrm{d}\lambda = \text{sech}(\lambda/2)/2\pi$. This derivation illustrates the process of obtaining $p(\lambda)$ from a noise schedule $\lambda(t)$. On the other hand, we can derive the noise schedule from the sampling probability of different noise intensities $p(\lambda)$. By integrating Equation 3, we have:

$$t = 1 - \int_{-\infty}^{\lambda} p(\lambda) \mathrm{d}\lambda = \mathcal{P}(\lambda), \tag{4}$$

$$\lambda = \mathcal{P}^{-1}(t), \tag{5}$$

where $\mathcal{P}(\lambda)$ represents the cumulative distribution function of $\lambda$. Thus we can obtain the noise schedule $\lambda$ by applying the inverse function $\mathcal{P}^{-1}$. In conclusion, during the training process, the importance sampling of varying noise intensities essentially equates to the modification of the noise schedules. To illustrate this concept, let's consider the Laplace distribution as an example , we can derive the cumulative distribution function $\mathcal{P}(\lambda) = 1 - \int \frac{1}{2b} \exp\left(-|\lambda - \mu|/b\right) \mathrm{d}\lambda = \frac{1}{2}\left(1 + \text{sgn}(\lambda - \mu)(1 - \exp(-|\lambda - \mu|/b))\right)$. Subsequently, we can obtain the inverse function to express the noise schedule in terms of $\lambda$: $\lambda = \mu - b\,\text{sgn}(0.5 - t)\ln(1 - 2|t - 0.5|)$. Here, $\text{sgn}(\cdot)$ denotes the signum function, which equals 1 for positive inputs, $-1$ for negative inputs. The pseudo-code for implementing the Laplace schedule in the training of diffusion models is presented in A.1.

This framework reveals that noise schedule design can be reframed as a probability distribution design problem. Rather than directly specifying how noise varies with time, we can instead focus on how to optimally distribute our sampling across different noise intensities. Our approach is also applicable to the recently popular flow matching with logit normal sampling scheme (Esser et al., 2024). Within our framework, we analyzed the distribution of its logSNR in A.4 and demonstrated its superiority over vanilla flow matching and cosine scheduling from the perspective of $p(\lambda)$.

| Noise Schedule | $p(\lambda)$ | $\lambda(t)$ |
|---|---|---|
| Cosine | $\operatorname{sech}(\lambda/2)/2\pi$ | $2\log\left(\cot\left(\frac{\pi t}{2}\right)\right)$ |
| Laplace | $e^{-\frac{|\lambda-\mu|}{b}}/2b$ | $\mu - b\operatorname{sgn}(0.5-t)\log(1-2|t-0.5|)$ |
| Cauchy | $\frac{1}{\pi}\frac{\gamma}{(\lambda-\mu)^2+\gamma^2}$ | $\mu + \gamma\tan\left(\frac{\pi}{2}(1-2t)\right)$ |
| Cosine Shifted | $\frac{1}{2\pi}\operatorname{sech}\left(\frac{\lambda-\mu}{2}\right)$ | $\mu + 2\log\left(\cot\left(\frac{\pi t}{2}\right)\right)$ |
| Cosine Scaled | $\frac{s}{2\pi}\operatorname{sech}\left(\frac{s\lambda}{2}\right)$ | $\frac{2}{s}\log\left(\cot\left(\frac{\pi t}{2}\right)\right)$ |

Table 1: Overview of various Noise Schedules. The table categorizes them into five distinct types: Cosine, Laplace, Cauchy, and two variations of Cosine schedules. The second column $p(\lambda)$ denotes the sampling probability at different noise intensities $\lambda$. The last column $\lambda(t)$ indicates how to sample noise intensities for training. We derived their relationship in Equation 3 and 5.

## 2.3 UNIFIED FORMULATION FOR DIFFUSION TRAINING

VDM++ (Kingma & Gao, 2023) proposes a unified formulation that encompasses recent prominent frameworks and loss weighting strategies for training diffusion models, as detailed below:

$$\mathcal{L}_w(\theta) = \frac{1}{2}\mathbb{E}_{\mathbf{x}\sim\mathcal{D},\boldsymbol{\epsilon}\sim\mathcal{N}(0,\mathbf{I}),\lambda\sim p(\lambda)}\left[\frac{w(\lambda)}{p(\lambda)}\left\|\hat{\boldsymbol{\epsilon}}_\theta(\mathbf{x}_\lambda;\lambda)-\boldsymbol{\epsilon}\right\|_2^2\right], \tag{6}$$

where $\mathcal{D}$ signifies the training dataset, noise $\boldsymbol{\epsilon}$ is drawn from a standard Gaussian distribution, and $p(\lambda)$ is the distribution of noise intensities. This formulation provides a flexible framework that can accommodate various diffusion training strategies. Different predicting targets, such as $\mathbf{x}_0$ and $\mathbf{v}$, can also be re-parameterized to $\boldsymbol{\epsilon}$-prediction. $w(\lambda)$ denotes the loss weighting strategy. Although adjusting $w(\lambda)$ is theoretically equivalent to altering $p(\lambda)$. In practical training, directly modifying $p(\lambda)$ to concentrate computational resources on training specific noise levels is more effective than enlarging the loss weight on specific noise levels. Given these insights, our research focuses on how to design an optimal $p(\lambda)$ that can effectively allocate computational resources across different noise levels. By carefully crafting the distribution of noise intensities, we aim to improve the overall training process and the quality of the resulting diffusion models. With the unified formulation providing a flexible framework for diffusion training, we can now apply these theoretical insights to practical settings. By carefully designing the distribution of noise intensities, we can optimize the training process and improve the performance of diffusion models in real-world applications. In the following section, we will explore practical strategies for noise schedules that leverage these insights to achieve better results.

## 2.4 PRACTICAL SETTINGS

Stable Diffusion 3 (Esser et al., 2024), EDM (Karras et al., 2022), and Min-SNR (Hang et al., 2023; Crowson et al., 2024) find that the denoising tasks with medium noise intensity is most critical to the overall performance of diffusion models. Therefore, we increase the probability of $p(\lambda)$ when $\lambda$ is of moderate size, and obtain a new noise schedule according to Section 2.2.

Specifically, we investigate four novel noise strategies, named Cosine Shifted, Cosine Scaled, Cauchy, and Laplace respectively. The detailed setting are listed in Table 1. Cosine Shifted use the hyperparameter $\mu$ to explore where the maximum probability should be used. Cosine Scaled explores how much the noise probability should be increased under the use of Cosine strategy to achieve better results. The Cauchy distribution, provides another form of function that can adjust both amplitude and offset simultaneously. The Laplace distribution is characterized by its mean $\mu$ and scale $b$, controls both the magnitude of the probability and the degree of concentration of the distribution. These strategies contain several hyperparameters, which we will explore in Section 3.5. Unless otherwise stated, we report the best hyperparameter results.

By re-allocating the computation resources at different noise intensities, we can train the complete denoising process. During sampling process, we align the sampled SNRs as the cosine schedule to ensure a fair comparison. Specifically, first we sample $\{t_0, t_1, \ldots, t_s\}$ from uniform distribution $\mathcal{U}[0, 1]$, then get the corresponding SNRs from Cosine schedule: $\{\frac{\alpha_{t_0}^2}{\sigma_{t_0}^2}, \frac{\alpha_{t_1}^2}{\sigma_{t_1}^2}, \ldots, \frac{\alpha_{t_s}^2}{\sigma_{t_s}^2}\}$. According

| Method | $w(\lambda)$ | $p(\lambda)$ |
|---|---|---|
| Cosine | $e^{-\lambda/2}$ | $\mathrm{sech}(\lambda/2)$ |
| Min-SNR | $e^{-\lambda/2} \cdot \min\{1, \gamma e^{-\lambda}\}$ | $\mathrm{sech}(\lambda/2)$ |
| Soft-Min-SNR | $e^{-\lambda/2} \cdot \gamma/(e^{\lambda} + \gamma)$ | $\mathrm{sech}(\lambda/2)$ |
| FM-OT | $(1 + e^{-\lambda})\mathrm{sech}^2(\lambda/4)$ | $\mathrm{sech}^2(\lambda/4)/8$ |
| EDM | $(1 + e^{-\lambda})(0.5^2 + e^{-\lambda})\mathcal{N}(\lambda; 2.4, 2.4^2)$ | $(0.5^2 + e^{-\lambda})\mathcal{N}(\lambda; 2.4, 2.4^2)$ |

Table 2: Comparison of different methods and related loss weighting strategies. The $w(\lambda)$ is introduced in Equation 6. The original $p(\lambda)$ for Soft-Min-SNR (Crowson et al., 2024) was developed within the EDM's denoiser framework. In this study, we align it with the cosine schedule to ensure a fair comparison.

to Equation 5, we get the corresponding $\{t'_0, t'_1, \ldots, t'_s\}$ by inverting these SNR values through the respective noise schedules. Finally, we use DDIM (Song et al., 2021) to sample with these new calculated $\{t'\}$. It is important to note that, from the perspective of the noise schedule, how to allocate the computation resource during inference is also worth reconsideration. We will not explore it in this paper and leave this as future work.

## 3 EXPERIMENTS

### 3.1 IMPLEMENTATION DETAILS

**Dataset.** We conduct experiments on ImageNet (Deng et al., 2009) with $256 \times 256$ and $512 \times 512$ resolution. For each image, we follow the preprocessing in Rombach et al. (2022) to center crop and encode images to latents. The resulting compressed latents have dimensions of $32 \times 32 \times 4$ for $256^2$ images and $64 \times 64 \times 4$ for $512^2$ images, effectively reducing the spatial dimensions while preserving essential visual information.

**Network Architecture.** We adopt DiT-B from Peebles & Xie (2023) as our backbone. We replace the last AdaLN Linear layer with vanilla linear. Others are kept the same as the original implementation. The patch size is set to 2 and the projected sequence length of $32 \times 32 \times 4$ is $\frac{32}{2} \cdot \frac{32}{2} = 256$. The class condition is injected through the adaptive layernorm. In this study, our primary objective is to demonstrate the effectiveness of our proposed noise schedule compared to existing schedules under a fixed training budget, rather than to achieve state-of-the-art results. Consequently, we do not apply our method to extra-large (XL) scale models.

**Training Settings.** We adopt the Adam optimizer (Kingma & Ba, 2014) with constant learning rate $1 \times 10^{-4}$. We set the batch size to 256 following Peebles & Xie (2023) and Gao et al. (2023). Each model is trained for 500K iterations (about 100 epochs) if not specified. Our implementation is primarily based on OpenDiT (Zhao et al., 2024) and experiments are mainly conducted on $8 \times 16$G V100 GPUs. Different from the default discrete diffusion setting with linear noise schedule in the code base, we implement the diffusion process in a continuous way. Specifically, we sample $t$ from uniform distribution $\mathcal{U}[0, 1]$.

**Baselines and Metrics.** We compare our proposed noise schedule with several baseline settings in Table 2. For each setting, we sample images using DDIM (Song et al., 2021) with 50 steps. Despite the noise strategy for different settings may be different, we ensure they share the same $\lambda = \log \text{SNR}$ at each sampling step. This approach is adopted to exclusively investigate the impact of the noise strategy during the training phase. Moreover, we report results with different classifier-free guidance scales(Ho & Salimans, 2021), and the FID is calculated using 10K generated images. We sample with three CFG scales and select the optimal one to better evaluate the actual performance of different models.

### 3.2 COMPARISON WITH BASELINE SCHEDULES AND LOSS WEIGHT DESIGNS

This section details the principal findings from our experiments on the ImageNet-256 dataset, focusing on the comparative effectiveness of various noise schedules and loss weightings in the context

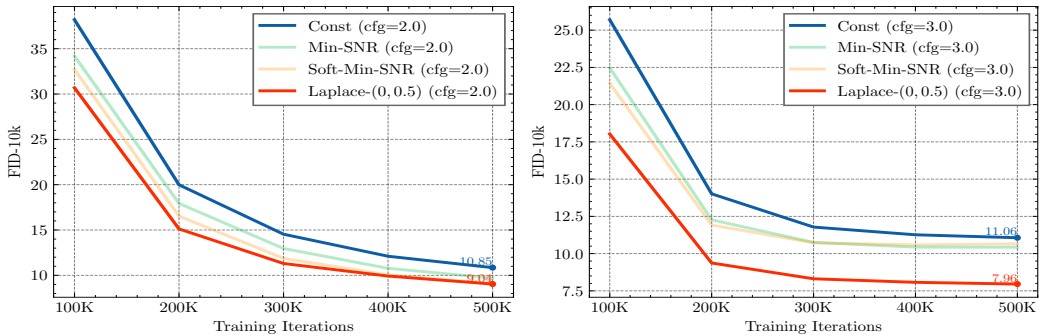

Figure 2: Comparison between adjusting the noise schedule, adjusting the loss weights and baseline setting. The Laplace noise schedule yields the best results and the fastest convergence speed.

of CFG values. Table 3 illustrates these comparisons, showcasing the performance of each method in terms of the FID-10K score.

The experiments reveal that our proposed noise schedules, particularly *Laplace*, achieve the most notable improvements over the traditional cosine schedule, as indicated by the bolded best scores and the blue numbers representing the reductions compared to baseline's best score of 10.85.

We also provide a comparison with methods that adjust the loss weight, including *Min-SNR* and *Soft-Min-SNR*. Unless otherwise specified, the hyperparameter $\gamma$ for both loss weighting schemes is set to 5. We find that although these methods can achieve better results than the baseline, they are still not as effective as our method of modifying the noise schedule. This indicates that deciding where to allocate more computational resources is more efficient than adjusting the loss weight. Compared with other noise schedules like EDM (Karras et al., 2022) and Flow Matching (Lipman et al., 2022), we found that no matter which CFG value, our results significantly surpass theirs under the same training iterations.

| Method | CFG=1.5 | CFG=2.0 | CFG=3.0 |
|---|---|---|---|
| Cosine (Nichol & Dhariwal, 2021) | 17.79 | 10.85 | 11.06 |
| EDM (Karras et al., 2022) | 26.11 | 15.09 | 11.56 |
| FM-OT (Lipman et al., 2022) | 24.49 | 14.66 | 11.98 |
| Min-SNR (Hang et al., 2023) | 16.06 | 9.70 | 10.43 |
| Soft-Min-SNR (Crowson et al., 2024) | 14.89 | 9.07 | 10.66 |
| Cosine Shifted (Hoogeboom et al., 2023) | 19.34 | 11.67 | 11.13 |
| Cosine Scaled | 12.74 | 8.04 | 11.02 |
| Cauchy | 12.91 | 8.14 | 11.02 |
| Laplace | 16.69 | 9.04 | ***7.96*** (-2.89) |

Table 3: Comparison of various noise schedules and loss weightings on ImageNet-256, showing the performance (in terms of FID-10K) of different methods under different CFG values. The best results highlighted in bold and the blue numbers represent the improvement when compared with the baseline FID 10.85. The line in gray is our suggested noise schedule.

Furthermore, we investigate the convergence speed of these method, and the results are shown in Figure 2. It can be seen that adjusting the noise schedule converges faster than adjusting the loss weight. Additionally, we also notice that the optimal training method may vary when using different CFG values for inference, but adjusting the noise schedule generally yields better results.

### 3.3 ROBUSTNESS ON DIFFERENT PREDICTING TARGETS

We evaluate the effectiveness of our designed noise schedule across three commonly adopted prediction targets: $\epsilon$, $\mathbf{x}_0$, and $\mathbf{v}$. The results are shown in Table 4.

We observed that regardless of the prediction target, our proposed Laplace strategy significantly outperforms the Cosine strategy. It's noteworthy that as the Laplace strategy focuses the computation on medium noise levels during training, the extensive noise levels are less trained, which could potentially affect the overall performance. Therefore, we have slightly modified the inference strategy of DDIM to start sampling from $t_{\max} = 0.99$.

| Predict Target | Noise Schedule | 100K | 200k | 300k | 400k | 500k |
|:---:|:---:|:---:|:---:|:---:|:---:|:---:|
| $\mathbf{x}_0$ | Cosine | 35.20 | 17.60 | 13.37 | 11.84 | 11.16 |
| | Laplace (Ours) | 21.78 | 10.86 | 9.44 | 8.73 | 8.48 |
| $\mathbf{v}$ | Cosine | 25.70 | 14.01 | 11.78 | 11.26 | 11.06 |
| | Laplace (Ours) | 18.03 | 9.37 | 8.31 | 8.07 | 7.96 |
| $\epsilon$ | Cosine | 28.63 | 15.80 | 12.49 | 11.14 | 10.46 |
| | Laplace (Ours) | 27.98 | 13.92 | 11.01 | 10.00 | 9.53 |

Table 4: Effectiveness evaluated using FID-10K score on different predicting targets: $\mathbf{x}_0$, $\epsilon$, and $\mathbf{v}$. The proposed *Laplace* schedule performs better than the baseline Cosine schedule along with training iterations.

### 3.4 ROBUSTNESS ON HIGH RESOLUTION IMAGES

To explore the robustness of the adjusted noise schedule to different resolutions, we also designed experiments on Imagenet-512. As pointed out by Chen (2023), the adding noise strategy will cause more severe signal leakage as the resolution increases. Therefore, we need to adjust the hyperparameters of the noise schedule according to the resolution.

Specifically, the baseline Cosine schedule achieves the best performance when the CFG value equals to 3. So we choose this CFG value for inference. Through systematic experimentation, we explored the appropriate values for the Laplace schedule's parameter $b$, testing within the range {0.5, 0.75, 1.0}, and determined that $b = 0.75$ was the most effective, resulting in an FID score of 9.09. This indicates that despite the need for hyperparameter tuning, adjusting the noise schedule can still stably bring performance improvements.

| Noise Schedule | Cosine | Laplace |
|:---:|:---:|:---:|
| FID-10K | 11.91 | ***9.09*** (-2.82) |

Table 5: FID-10K results on ImageNet-512. All models are trained for 500K iterations.

### 3.5 ABLATION STUDY

We conduct an ablation study to analyze the impact of hyperparameters on various distributions of $p(\lambda)$, which are enumerated below.

**Laplace** distribution, known for its simplicity and exponential decay from the center, is straightforward to implement. We leverage its symmetric nature and adjust the scale parameter to center the peak at the middle timestep. We conduct experiments with different Laplace distribution scales $b \in \{0.25, 0.5, 1.0, 2.0, 3.0\}$. The results are shown in Figure 3. The baseline with standard cosine schedule achieves FID score of 17.79 with CFG=1.5, 10.85 with CFG=2.0, and 11.06 with CFG=3.0 after 500K iterations. We can see that the model with Laplace distribution scale $b = 0.5$ achieves the best performance 7.96 with CFG=3.0, which is relatively **26.6%** better than the baseline.

**Cauchy** distribution is another heavy-tailed distribution that can be used for noise schedule design. The distribution is not symmetric when the location parameter is not 0. We conduct experiments with different Cauchy distribution parameters and the results are shown in Table 6. Cauchy(0, 0.5) means $\frac{1}{\pi} \frac{\gamma}{(\lambda-\mu)^2+\gamma^2}$ with $\mu = 0, \gamma = 0.5$. We can see that the model with $\mu = 0$ achieve better performance than the other two settings when fixing $\gamma$ to 1. It means that the model with more probability mass around $\lambda = 0$ performs better than others biased to negative or positive directions.

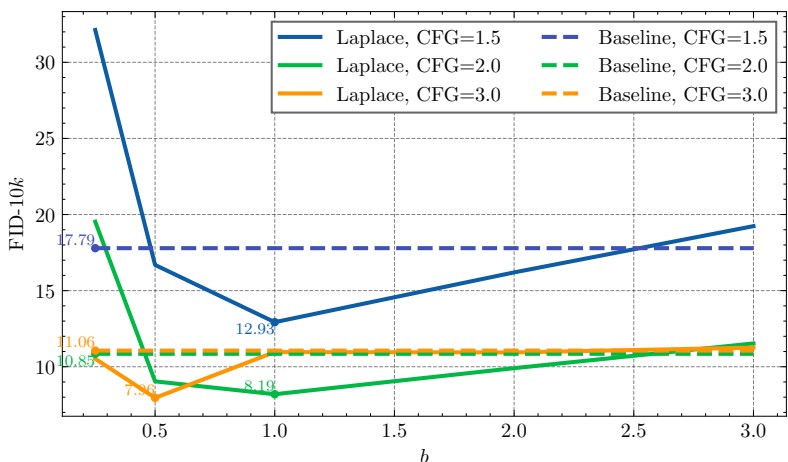

Figure 3: FID-10K results on ImageNet-256 with location parameter $\mu$ fixed to 0 and different Laplace distribution scales $b$ in $\{0.25, 0.5, 1.0, 2.0, 3.0\}$. Baseline denotes standard cosine schedule.

|  | Cauchy(0, 0.5) | Cauchy(0, 1) | Cauchy(-1, 1) | Cauchy(1, 1) |
|---|---|---|---|---|
| CFG=1.5 | 12.91 | 14.32 | 18.12 | 16.60 |
| CFG=2.0 | **8.14** | 8.93 | 10.38 | 10.19 |
| CFG=3.0 | 11.02 | 11.26 | 10.81 | 10.94 |

Table 6: FID-10k results on ImageNet-256 with different Cauchy distribution parameters.

**Cosine Shifted** (Hoogeboom et al., 2023) is the shifted version of the standard cosine schedule. We evaluate the schedules with both positive and negative $\mu$ values to comprehensively assess its impact on model performance. Shifted with $\mu = 1$ achieves FID-10k score $\{19.34, 11.67, 11.13\}$ with CFG $\{1.5, 2.0, 3.0\}$. Results with shifted value $\mu = -1$ are $\{19.30, 11.48, 11.28\}$. Comparatively, both scenarios demonstrate inferior performance relative to the baseline cosine schedule ($\mu = 0$). Additionally, by examining the data presented in Table 6, we find concentrated on $\lambda = 0$ can best improve the results.

**Cosine Scaled** is also a modification of Cosine schedule. When $s = 1$, it becomes the standard Cosine version. $s > 1$ means sampling more heavily around $\lambda = 0$ while $s < 1$ means sampling more uniformly of all $\lambda$. We report related results in Table 7. Our experimental results reveal a clear trend: larger values of $s(s > 1)$ consistently outperform the baseline, highlighting the benefits of focused sampling near $\lambda = 0$. However, it's crucial to note that $s$ should not be excessively large and must remain within a valid range to maintain stable training dynamics. For example, decreasing $1/s$ from 0.5 to 0.25 hurts the performance and cause the FID score to drop. Striking the right balance is key to optimizing performance. In our experiments, a model trained with $s = 2$ achieved a remarkable score of 8.04, representing a substantial 25.9% improvement over the baseline.

The experiments with various noise schedules, including Laplace, Cauchy, Cosine Shifted, and Cosine Scaled, reveal a shared phenomenon: *models perform better when the noise distribution or schedule is concentrated around $\lambda = 0$.* For the Laplace distribution, a scale of $b = 0.5$ yielded the best performance, outperforming the baseline by 26.6%. In the case of the Cauchy distribution, models with a location parameter $\mu = 0$ performed better than those with $\mu$ values biased towards negative or positive directions. The Cosine Shifted schedule showed inferior performance when shifted away from $\mu = 0$, while the Cosine Scaled schedule demonstrated that larger values of $s$ (sampling more heavily around $\lambda = 0$) consistently outperformed the baseline, with an optimal improvement of 25.9% at $s = 2$. This consistent trend suggests that focusing the noise distribution or schedule near $\lambda = 0$ is beneficial for model performance. While these different schedules take various mathematical forms, they all achieve similar optimal performance when given equivalent training budgets. The specific mathematical formulation is less crucial than the underlying design philosophy: increasing the sampling probability of intermediate noise levels. This principle provides a simple yet effective guideline for designing noise schedules.

| $1/s$ | 1.3 | 1.1 | 0.5 | 0.25 |
|---|---|---|---|---|
| CFG=1.5 | 39.74 | 22.60 | 12.74 | 15.83 |
| CFG=2.0 | 23.38 | 12.98 | **8.04** | 8.64 |
| CFG=3.0 | 13.94 | 11.16 | 11.02 | 8.26 |

Table 7: FID-10k results on ImageNet-256 with different scales of Cosine Scaled distribution.

## 4 RELATED WORKS

### EFFICIENT DIFFUSION TRAINING

Generally speaking, the diffusion model uses a network with shared parameters to denoise different noise intensities. However, the different noise levels may introduce conflicts during training, which makes the convergence slow. P2 (Choi et al., 2022) improves image generation performance by prioritizing the learning of perceptually rich visual concepts during training through a redesigned weighting scheme. Min-SNR (Hang et al., 2023) seeks the Pareto optimal direction for different tasks, achieves better convergence on different predicting targets. HDiT (Crowson et al., 2024) propose a soft version of Min-SNR to further improve the efficiency on high resolution image synthesis. Stable Diffusion 3 (Esser et al., 2024) puts more sampling weight on the middle timesteps by multiplying the distribution of logit normal distribution.

On the other hand, architecture modification is also explored to improve diffusion training. DiT (Peebles & Xie, 2023) proposes adaptive Layer Normalization with zero initialization to improve the training of Transformer architectures. Building upon this design, MM-DiT (Esser et al., 2024) extends the approach to a multi-modal framework (text to image) by incorporating separate sets of weights for each modality. HDiT (Crowson et al., 2024) uses a hierarchical transformer structure for efficient, linear-scaling, high-resolution image generation. A more robust ADM UNet with better training dynamics is proposed in EDM2 (Karras et al., 2024) by preserving activation, weight, and update magnitudes. In this work, we directly adopt the design from DiT (Peebles & Xie, 2023) and focus on investigating the importance sampling schedule in diffusion models.

### NOISE SCHEDULE DESIGN FOR DIFFUSION MODELS

The design of the noise schedule plays a critical role in training diffusion models. In DDPM, Ho et al. (2020) propose linear schedule for the noise level, which was later adopted by Stable Diffusion (Rombach et al., 2022) version 1.5 and 2.0. However, the linear noise schedule introduces signal leakage at the highest noise step (Lin et al., 2024; Tang et al., 2023), hindering performance when sampling starts from a Gaussian distribution. Improved DDPM (Nichol & Dhariwal, 2021) introduces a cosine schedule aimed at bringing the sample with the highest noise level closer to pure Gaussian noise. EDM (Karras et al., 2022) proposes a new continuous framework and make the logarithm of noise intensity sampled from a Gaussian distribution. Flow matching with optimal transport (Lipman et al., 2022; Liu et al., 2022) linearly interpolates the noise and data point as the input of flow-based models. Chen (2023) underscored the need for adapting the noise schedule according to the image resolution. Hoogeboom et al. (2023) found that cosine schedule exhibits superior performance for images of $32 \times 32$ and $64 \times 64$ resolutions and propose to shift the cosine schedule to train on images with higher resolutions.

## 5 CONCLUSION

In this paper, we present a novel method for enhancing the training of diffusion models by strategically redefining the noise schedule. Our theoretical analysis demonstrates that this approach is equivalent to performing importance sampling on the noise. Empirical results show that our proposed Laplace noise schedule, which focuses computational resources on mid-range noise levels, yields superior performance compared to adjusting loss weights under constrained computational budgets. This study not only contributes significantly to the development of efficient training techniques for diffusion models but also offers promising potential for future large-scale applications.

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

# A APPENDIX

## A.1 DETAILED IMPLEMENTATION FOR NOISE SCHEDULE

We provide a simple PyTorch implementation for the Laplace noise schedule and its application in training. This example can be adapted to other noise schedules, such as the Cauchy distribution, by replacing the `laplace_noise_schedule` function. The model accepts noisy samples $\mathbf{x}_t$, timestep $t$, and an optional condition tensor $\mathbf{c}$ as inputs. This implementation supports prediction of $\{\mathbf{x}_0, \mathbf{v}, \boldsymbol{\epsilon}\}$.

```python
import torch

def laplace_noise_schedule(mu=0.0, b=0.5):
    # refer to Table 1
    lmb = lambda t: mu - b * torch.sign(0.5 - t) * \
        torch.log(1 - 2 * torch.abs(0.5 - t))
    snr_func   = lambda t: torch.exp(lmb(t))
    alpha_func = lambda t: torch.sqrt(snr_func(t) / (1 + snr_func(t)))
    sigma_func = lambda t: torch.sqrt(1 / (1 + snr_func(t)))

    return alpha_func, sigma_func

def training_losses(model, x, timestep, condition, noise=None,
                    predict_target="v", mu=0.0, b=0.5):

    if noise is None:
        noise = torch.randn_like(x)

    alpha_func, sigma_func = laplace_noise_schedule(mu, b)
    alphas = alpha_func(timestep)
    sigmas = sigma_func(timestep)

    # add noise to sample
    x_t = alphas.view(-1, 1, 1, 1) * x + sigmas.view(-1, 1, 1, 1) * noise
    # velocity
    v_t = alphas.view(-1, 1, 1, 1) * noise - sigmas.view(-1, 1, 1, 1) * x

    model_output = model(x_t, timestep, condition)
    if predict_target == "v":
        loss = (v_t - model_output) ** 2
    elif predict_target == "x0":
        loss = (x - model_output) ** 2
    else: # predict_target == "noise":
        loss = (noise - model_output) ** 2

    return loss.mean()
```

## A.2 DETAILS FOR PROPOSED LAPLACE AND CAUCHY DESIGN

For a Laplace distribution with location parameter $\mu$ and scale parameter $b$, the probability density function (PDF) is given by:

$$p(\lambda) = \frac{1}{2b} \exp\left(-\frac{|\lambda - \mu|}{b}\right) \tag{7}$$

The cumulative distribution function (CDF) can be derived as follows:

$$1 - t = \int_{-\infty}^{\lambda} p(x)\, \mathrm{d}x$$

$$= \int_{-\infty}^{\lambda} \frac{1}{2b} \exp\left(-\frac{|x - \mu|}{b}\right)\, \mathrm{d}x$$

$$= \frac{1}{2}\left(1 + \mathrm{sgn}(\lambda - \mu)\left(1 - \exp\left(-\frac{|\lambda - \mu|}{b}\right)\right)\right)$$

To obtain $\lambda$ as a function of $t$, we solve the inverse function:

$$\lambda = \mu - b\,\mathrm{sgn}(0.5 - t)\ln(1 - 2|t - 0.5|)$$

For a Cauchy distribution with location parameter $\mu$ and scale parameter $\gamma$, the PDF is given by:

$$f(\lambda; \mu, \gamma) = \frac{1}{\pi\gamma}\left[1 + \left(\frac{\lambda - \mu}{\gamma}\right)^2\right]^{-1} \tag{8}$$

The corresponding CDF is:

$$F(\lambda; \mu, \gamma) = \frac{1}{2} + \frac{1}{\pi}\arctan\left(\frac{\lambda - \mu}{\gamma}\right) \tag{9}$$

To derive $\lambda(t)$, we proceed as follows:

$$1 - t = F(\lambda; \mu, \gamma) \tag{10}$$

$$1 - t = \frac{1}{2} + \frac{1}{\pi}\arctan\left(\frac{\lambda - \mu}{\gamma}\right) \tag{11}$$

$$t = \frac{1}{2} - \frac{1}{\pi}\arctan\left(\frac{\lambda - \mu}{\gamma}\right) \tag{12}$$

Solving for $\lambda$, we obtain:

$$\lambda(t) = \mu + \gamma\tan\left(\frac{\pi}{2}(1 - 2t)\right) \tag{13}$$

### A.3 COMBINATION BETWEEN NOISE SCHEDULE AND TIMESTEP IMPORTANCE SAMPLING

We observe that incorporating importance sampling of timesteps into the cosine schedule bears similarities to the Laplace schedule. Typically, the distribution of timestep $t$ is uniform $\mathcal{U}[0, 1]$. To increase the sampling frequency of middle-level timesteps, we propose modifying the sampling distribution to a simple polynomial function:

$$p(t') = \begin{cases} C \cdot t'^n, & t' < \frac{1}{2} \\ C \cdot (1 - t')^n, & t' \geq \frac{1}{2}, \end{cases} \tag{14}$$

where $C = (n + 1)2^n$ is the normalization factor ensuring that the cumulative distribution function (CDF) equals 1 at $t = 1$.

To sample from this distribution, we first sample $t$ uniformly from $(0, 1)$ and then map it using the following function:

$$t' = \begin{cases} \left(\frac{1}{2}\right)^{\frac{n}{n+1}} t^{\frac{1}{n+1}}, & t < \frac{1}{2} \\ 1 - \left(\frac{1}{2}\right)^{\frac{n}{n+1}} (1 - t)^{\frac{1}{n+1}}, & t \geq \frac{1}{2}, \end{cases} \tag{15}$$

We incorporate the polynomial sampling of $t$ into the cosine schedule $\lambda = -2 \log \tan \frac{\pi t}{2}$, whose inverse function is $t = \frac{2}{\pi} \arctan \exp\left(-\frac{\lambda}{2}\right)$. Let us first consider the situation where $t < \frac{1}{2}$:

$$\left(\frac{1}{2}\right)^{\frac{-n}{n+1}} t^{\frac{1}{n+1}} = \frac{2}{\pi} \arctan \exp\left(-\frac{\lambda}{2}\right) \tag{16}$$

$$t = 2^n \left(\frac{2}{\pi} \arctan \exp\left(-\frac{\lambda}{2}\right)\right)^{n+1} \tag{17}$$

We then derive the expression with respect to $d\lambda$:

$$\frac{dt}{d\lambda} = 2^n \left(\frac{2}{\pi}\right)^{n+1} (n+1) \left(\arctan \exp\left(-\frac{\lambda}{2}\right)\right)^n \frac{1}{1 + \exp(-\lambda)} \frac{1}{-2} \exp(-\lambda/2) \tag{18}$$

$$p(\lambda) = (n+1) \frac{4^n}{\pi^{(n+1)}} \arctan^n \exp\left(-\frac{\lambda}{2}\right) \frac{\exp(-\frac{1}{2}\lambda)}{1 + \exp(-\lambda)} \tag{19}$$

$$\tag{20}$$

Considering symmetry, we obtain the final distribution with respect to $\lambda$ as follows:

$$p(\lambda) = (n+1) \frac{4^n}{\pi^{(n+1)}} \arctan^n \exp\left(-\frac{|\lambda|}{2}\right) \frac{\exp(-\frac{1}{2}|\lambda|)}{1 + \exp(-|\lambda|)} \tag{21}$$

We visualize the schedule discussed above and compare it with Laplace schedule in Figure 4. We can see that $b = 1$ for Laplace and $n = 2$ for cosine-ply matches well. We also conduct experiments on such schedule and present results in Table 8. They perform similar and both better than the standard cosine schedule.

We visualize the schedules discussed above and compare them with the Laplace schedule in Figure 4. The results demonstrate that Laplace with $b = 1$ and cosine-ply with $n = 2$ exhibit a close correspondence. To evaluate the performance of these schedules, we conducted experiments and present the results in Table 8. Both the Laplace and cosine-ply schedules show similar performance, and both outperform the standard cosine schedule.

| Iterations | 100,000 | 200,000 | 300,000 | 400,000 | 500,000 |
|---|---|---|---|---|---|
| Cosine-ply ($n = 2$) | 28.65 | 13.77 | 10.06 | 8.69 | 7.98 |
| Laplace ($b = 1$) | 28.89 | 13.90 | 10.17 | 8.85 | 8.19 |

Table 8: Performance comparison of cosine-ply ($n = 2$) and Laplace ($\mu = 1$) schedules over different iteration counts

## A.4 FLOW MATCHING WITH LOGIT-NORMAL SAMPLING

In Stable Diffusion 3 (Esser et al., 2024) and Movie Gen (Polyak et al., 2024), logit-normal sampling is applied to improve the training efficiency of flow models. To better understand this approach, we present a detailed derivation from the logit-normal distribution to the probability density function of logSNR $\lambda$.

Let the Logit transformation $X = \text{logit}(t)$ of random variable $t$ follow a normal distribution:

$$X \sim \mathcal{N}(\mu, \sigma^2) \tag{22}$$

Then, the probability density function of $t$ is:

$$p(t; \mu, \sigma) = \frac{1}{\sigma \cdot t \cdot (1 - t) \cdot \sqrt{2\pi}} \exp\left(-\frac{(\text{logit}(t) - \mu)^2}{2\sigma^2}\right), \quad t \in (0, 1) \tag{23}$$

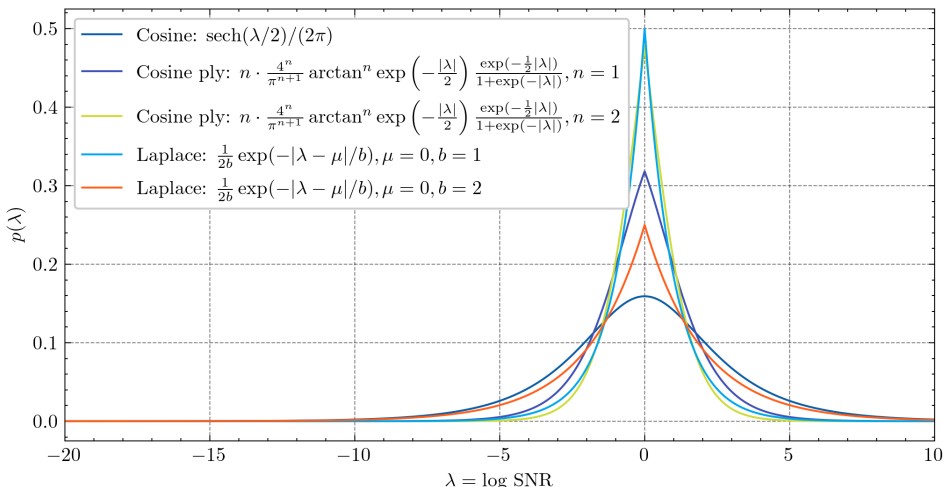

Figure 4: Visualization of $p(\lambda)$ for Laplace schedule and cosine schedule with polynomial timestep sampling.

where $\text{logit}(t) = \log\left(\frac{t}{1-t}\right)$, and $\mu$ and $\sigma$ are constants.

Consider the variable transformation:

$$\lambda = 2\log\left(\frac{1-t}{t}\right) \tag{24}$$

Our goal is to find the probability density function $p(\lambda)$ of random variable $\lambda$.

First, we solve for $t$ in terms of $\lambda$:

$$\frac{\lambda}{2} = \log\left(\frac{1-t}{t}\right)$$

$$e^{\frac{\lambda}{2}} = \frac{1-t}{t}$$

$$1 - t = te^{\frac{\lambda}{2}}$$

$$1 = t\left(1 + e^{\frac{\lambda}{2}}\right)$$

$$t(\lambda) = \frac{1}{1 + e^{\frac{\lambda}{2}}}$$

Next, we calculate the Jacobian determinant $\left|\frac{\mathrm{d}t}{\mathrm{d}\lambda}\right|$:

$$t(\lambda) = \frac{1}{1 + e^{\frac{\lambda}{2}}}$$

$$\frac{\mathrm{d}t}{\mathrm{d}\lambda} = -\frac{e^{\frac{\lambda}{2}} \cdot \frac{1}{2}}{(1 + e^{\frac{\lambda}{2}})^2}$$

$$\left|\frac{\mathrm{d}t}{\mathrm{d}\lambda}\right| = \frac{e^{\frac{\lambda}{2}}}{2(1 + e^{\frac{\lambda}{2}})^2}$$

Using the variable transformation formula:

$$p(\lambda) = p(t(\lambda); \mu, \sigma) \cdot \left|\frac{\mathrm{d}t}{\mathrm{d}\lambda}\right| \tag{25}$$

We calculate $p(t(\lambda); \mu, \sigma)$:

$$\text{logit}(t(\lambda)) = \log\left(\frac{t(\lambda)}{1 - t(\lambda)}\right) = \log\left(\frac{\frac{1}{1+e^{\frac{\lambda}{2}}}}{\frac{e^{\frac{\lambda}{2}}}{1+e^{\frac{\lambda}{2}}}}\right) = -\frac{\lambda}{2}$$

$$p(t(\lambda); \mu, \sigma) = \frac{(1 + e^{\frac{\lambda}{2}})^2}{\sigma e^{\frac{\lambda}{2}}\sqrt{2\pi}} \exp\left(-\frac{(\mu + \frac{\lambda}{2})^2}{2\sigma^2}\right)$$

Multiplying by the Jacobian determinant:

$$p(\lambda) = \frac{(1 + e^{\frac{\lambda}{2}})^2}{\sigma e^{\frac{\lambda}{2}}\sqrt{2\pi}} \exp\left(-\frac{(\mu + \frac{\lambda}{2})^2}{2\sigma^2}\right) \cdot \frac{e^{\frac{\lambda}{2}}}{2(1 + e^{\frac{\lambda}{2}})^2}$$

$$= \frac{1}{2\sigma\sqrt{2\pi}} \exp\left(-\frac{(\lambda + 2\mu)^2}{8\sigma^2}\right)$$

Therefore, the probability density function of $\lambda$ is:

$$p(\lambda) = \frac{1}{2\sigma\sqrt{2\pi}} \exp\left(-\frac{(\lambda + 2\mu)^2}{8\sigma^2}\right), \quad \lambda \in (-\infty, +\infty) \tag{26}$$

This shows that $\lambda$ follows a normal distribution with mean $-2\mu$ and variance $4\sigma^2$:

$$\lambda \sim \mathcal{N}(-2\mu, 4\sigma^2) \tag{27}$$

The mean and variance are:

$$\mathbb{E}[\lambda] = -2\mu$$

$$\text{Var}(\lambda) = 4\sigma^2$$

To verify normalization, we integrate $p(\lambda)$ over its domain:

$$\int_{-\infty}^{+\infty} p(\lambda)\,d\lambda = \int_{-\infty}^{+\infty} \frac{1}{2\sigma\sqrt{2\pi}} \exp\left(-\frac{(\lambda + 2\mu)^2}{8\sigma^2}\right) d\lambda$$

$$\text{Let } z = \frac{\lambda + 2\mu}{2\sqrt{2}\sigma} \Rightarrow d\lambda = 2\sqrt{2}\sigma\,dz$$

$$= \frac{2\sqrt{2}\sigma}{2\sigma\sqrt{2\pi}} \int_{-\infty}^{+\infty} e^{-z^2}\,dz$$

$$= \frac{1}{\sqrt{\pi}} \cdot \sqrt{\pi} = 1$$

Thus, $p(\lambda)$ satisfies the normalization condition for probability density functions.

We compare the standard cosine scheudle (Nichol & Dhariwal, 2021), Flow Matching (Liu et al., 2022; Lipman et al., 2022), and Flow Matching with Logit-normal sampling (Esser et al., 2024; Polyak et al., 2024). The probability density functions of these schedules are visualized in Figure 5. Our analysis reveals that Flow Matching with Logit-normal sampling concentrates more probability mass around $\lambda = 0$ compared to both the standard Cosine and Flow Matching schedules, resulting in improved training efficiency (Esser et al., 2024; Polyak et al., 2024).

## A.5 IMPORTANCE OF TIME INTERVALS

To investigate the significance of training intervals, we conducted controlled experiments using a simplified setup. We divided the time range $(0, 1)$ into four equal segments: $\text{bin}_i = \left(\frac{i}{4}, \frac{i+1}{4}\right), i = 0, 1, 2, 3$. We first trained a base model $\mathbf{M}$ over the complete range $(0, 1)$ for 1M iterations, then

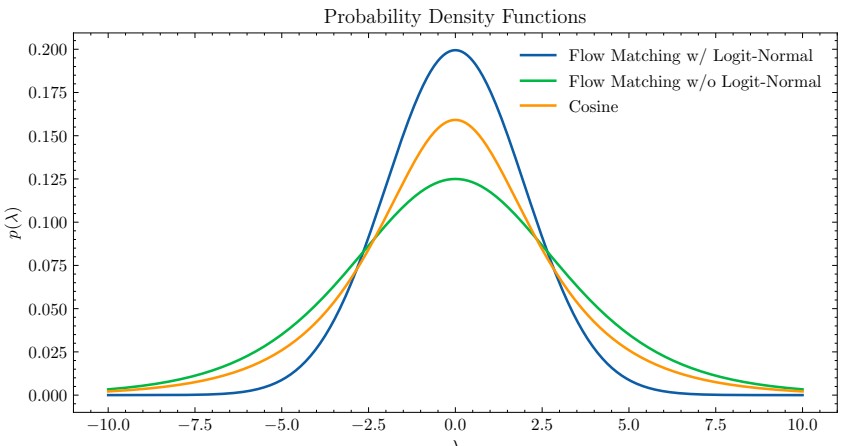

Figure 5: Comparison of probability density functions for different flow matching approaches. The plot shows three distributions: Flow Matching with Logit-Normal sampling (blue), Flow Matching without Logit-Normal sampling (green), and the Cosine schedule (orange).

fine-tuned it separately on each bin for 140k iterations to obtain four specialized checkpoints $\mathbf{m}_i$, $i = 0, 1, 2, 3$.

For evaluation, we designed experiments using both the base model $\mathbf{M}$ and fine-tuned checkpoints $\mathbf{m}_i$. To assess the importance of each temporal segment, we selectively employed the corresponding fine-tuned checkpoint during its specific interval while maintaining the base model for remaining intervals. For example, when evaluating $\text{bin}_0$, we used $\mathbf{m}_0$ within its designated interval and $\mathbf{M}$ elsewhere.

The FID results across these four experimental configurations are presented in Figure 6. Our analysis reveals that optimizing intermediate timesteps (bin1 and bin2) yields superior performance, suggesting the critical importance of these temporal regions in the diffusion process.

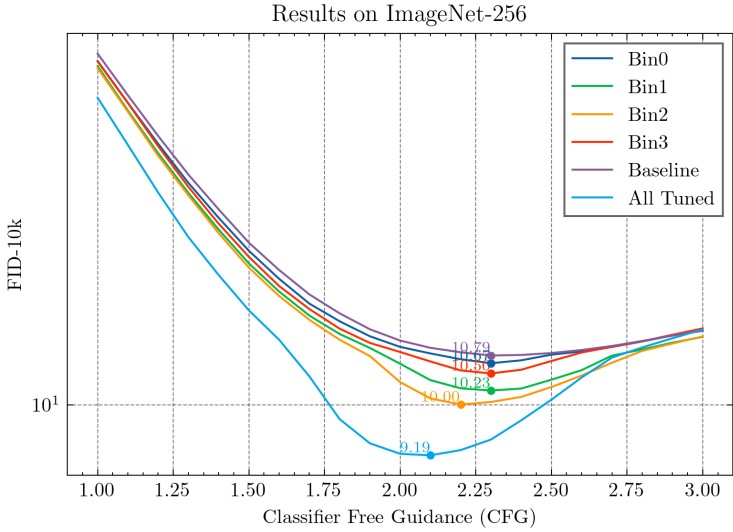

Figure 6: Comparative analysis of interval-specific fine-tuning effects. When sampling within interval $\left(\frac{1}{4}, \frac{2}{4}\right)$, "Bin1" indicates the use of fine-tuned weights $\mathbf{m}_1$, while $\mathbf{M}$ is used for other intervals. "Baseline" represents the use of base model $\mathbf{M}$ throughout all intervals, and "All Tuned" denotes the application of interval-specific fine-tuned models within their respective ranges.

## A.6 IMPORTANCE SAMPLING AS LOSS WEIGHT

We investigate the comparative effectiveness of our approach when applied as a noise schedule versus a loss weighting mechanism. We adopt Equation 21 as our primary noise schedule due to its foundation in the cosine schedule and demonstrated superior FID performance. To evaluate its versatility, we reformulate the importance sampling as a loss weighting strategy and compare it against established weighting schemes, including Min-SNR and Soft-Min-SNR.

|          | Cosine | Cosine-Ply ($n$=2) | Min-SNR | Soft-Min-SNR | Cosine-Ply as weight |
|----------|--------|---------------------|---------|--------------|----------------------|
| FID-10K  | 10.85  | 7.98                | 9.70    | 9.07         | 8.88                 |

Table 9: Quantitative comparison of different noise scheduling strategies and loss weighting schemes. Lower FID scores indicate better performance.

Figure 7 illustrates the loss weight derived from Cosine-Ply ($n$=2) schedule alongside Min-SNR and Soft-Min-SNR. We can observe that under the setting of predict target as $\mathbf{v}$, Min-SNR and Soft-Min-SNR can be seemed as putting more weight on intermediate levels, aligning with our earlier findings on the importance of middle-level noise densities.

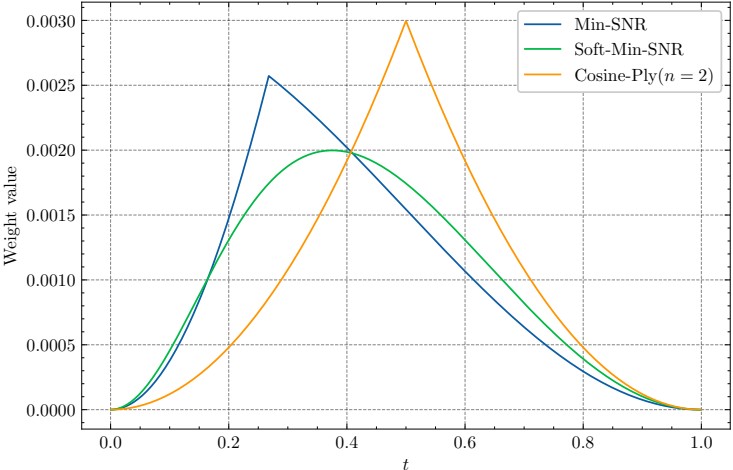

Figure 7: Visualization of different loss weight schemes.

## A.7 ADDITIONAL EXPERIMENTS ON OTHER DATASETS

ImageNet, comprising over one million natural images, has been widely adopted as a benchmark dataset for validating improvements in diffusion models (Peebles & Xie, 2023; Karras et al., 2024).

In addition to ImageNet, we evaluate our approach on the CelebA (Liu et al., 2015) dataset ($64 \times 64$ resolution in pixel space), which consists of face images. We employ a DiT architecture (12 layers, embedding dimension of 512, 8 attention heads, and patch size of 4) using different noise schedules. This is an unconditional generation setting within a single domain. We present FID results as follows:

| FID ↓         | 100k    | 150k    |
|---------------|---------|---------|
| cosine        | 10.0696 | 7.93795 |
| Laplace (ours) | 7.93795 | 6.58359 |

Table 10: FID scores on CelebA dataset at different training iterations

We also follow Stable Diffusion 3 (Esser et al., 2024), train on a more complicated dataset CC12M (Changpinyo et al., 2021) dataset (over 12M image-text pairs) and report the FID results

here. We download the dataset using `webdataset`. We train a DiT-base model using CLIP as text conditioner. The images are cropped and resized to $256 \times 256$ resolution, compressed to $32 \times 32 \times 4$ latents and trained for 200k iterations at batch size 256.

| FID ↓ | 200k |
|---|---|
| cosine | 58.3619 |
| Laplace (ours) | 54.3492 (-4.0127) |

Table 11: FID scores on CC12M dataset at 200k iterations

Our method demonstrated strong generalization capabilities across both unconditional image generation using the CelebA dataset and text-to-image generation using the CC12M dataset.

## A.8 ADDITIONAL VISUAL RESULTS

We present addition visual results in Figure 8 to demonstrate the differences in generation quality between models trained with Cosine and our proposed Laplace schedule. Each case presents two rows of outputs, where the upper row shows results from the cosine schedule and the lower row displays results from our Laplace schedule. Each row contains five images corresponding to models trained for 100k, 200k, 300k, 400k, and 500k iterations, illustrating the progression of generation quality across different training stages. For each case, the initial noise inputs are identical. As shown in the results, our method achieves faster convergence in both basic object formation (at 100k iterations) and fine detail refinement, demonstrating superior learning efficiency throughout the training process.

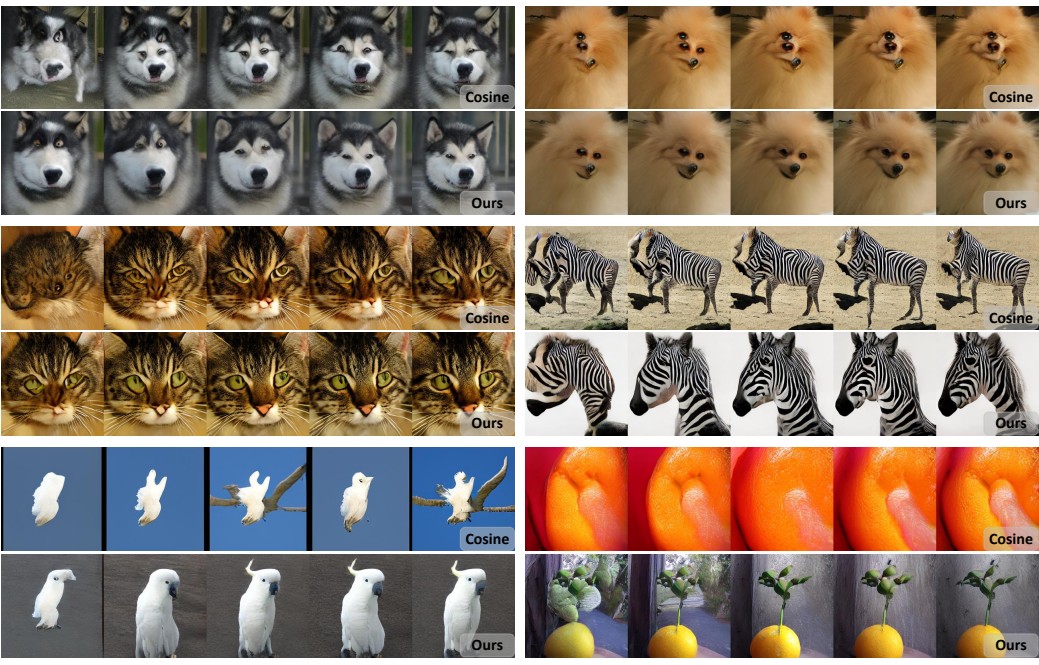

Figure 8: Visual comparison of results generated by model trained by cosine schedule and our proposed Laplace. For each case, the above row is generated by cosine schedule, the below is generated by Laplace. The 5 images from left to right represents the results generated by the model trained for 100k, 200k, 300k, 400k, and 500k iterations.

