# OpenReview forum: "Improved Noise Schedule for Diffusion Training"
_ICLR.cc/2025/Conference — Submitted to ICLR 2025_

### Official Review · Reviewer_f1Vg · 2024-10-24

**Soundness:** 3
**Presentation:** 2
**Contribution:** 2
**Rating:** 5
**Confidence:** 2

**Summary:**

This paper presents a modified noise schedule that increases the sampling frequency around $\log SNR = 0$ to accelerate convergence and generate robust, accurate predictions in diffusion models. Experimental results demonstrate the effectiveness of this approach and its advantages across various prediction targets in the ImageNet benchmark.

**Strengths:**

S1. The problem of improving the training efficiency of diffusion models is well-motivated because diffusion models are known to be computationally intensive.

S1. The proposed method redefines the noise schedule to enhance the training of diffusion models, resulting in higher efficiency and demonstrating strong predictive performance.

S2. The effectiveness of the proposed method is confirmed through extensive experiments.

**Weaknesses:**

W1. The theoretical support for the proposed method is lacking. While the paper establishes a connection between the sampling probability and the noise schedule, the conclusion regarding the choice of λ at a moderate strength primarily draws from prior work, lacking unique insights or innovations in its key findings.

W2. The experimental results' persuasive power is insufficient; additional datasets, such as CIFAR-10 and CelebA, are needed to validate the generation performance. Increasing the number of generated images for FID calculation would also be beneficial.

W3. Minor presentation issues. For instance, Table 1 differs in formatting from other tables.

**Questions:**

Q1. Is there more empirical evidence to support the claim that "In practical training, directly modifying p(λ) to concentrate computational resources on specific noise levels is more effective than increasing the loss weight on those levels"?

Q2. The paper mentions calculating FID using 10K generated images. Is this sample size sufficient, and can the FID calculation be based on other widely adopted datasets such as CIFAR-10 and CelebA?

Q3. The paper uses only FID to indicate the impact on quality. Could visual examples be provided to demonstrate that this sampling method does not adversely affect the generated outputs?

---

> ### Author Response · Authors · 2024-11-21
>
> Thank you for your thoughtful feedback. We address your questions or concerns below.
>
> ***Q1: Effectiveness of noise schedule design over loss weighting schemes***
>
> We present additional experimental comparisons between our noise scheduling approach and its derived loss weighting strategy. For this analysis, we employ the Cosine-Ply ($n=2$) schedule described in Section A.3, which builds upon the conventional Cosine schedule and demonstrates superior FID performance. While implementing Cosine-Ply as a loss weighting mechanism yields improvements over the baseline, its effectiveness remains secondary to its direct application as a noise schedule. We also include related results in revised A.6.
>
> |         | Cosine (baseline) | Cosine-Ply (n=2) | Cosine-Ply as weight |
> | ------- | ----------------- | ---------------- | -------------------- |
> | FID-10K | 10.85             | 7.98             | 8.88                 |
>
> Under the prediction target $\mathbf{v}$, Min-SNR and Soft-Min-SNR employ loss weighting functions of $\frac{\min \{ \text{SNR}, \gamma \} }{\text{SNR} + 1}$ and $\frac{1}{\text{SNR} + 1}\frac{1}{\text{SNR}^{-1} + \gamma^{-1}}$, respectively. While these approaches emphasize intermediate noise densities through weighting (shown in Figure 7 in the revised PDF), our proposed method, which directly increases sampling frequency in these regions, demonstrates superior performance compared to such weighting-based strategies.
>
>
> ***Q2 Sufficiency of 10k samples to calculate fid scores and more results on other datasets***
>
> The use of 10K samples for FID calculation follows established practices in recent literature, such as U-ViT [1]. It is sufficient to use FID-10k to validate the effectiveness of proposed method. Increasing sample size from 10K to 50K leads to lower FID. To further validate our results, we extended our evaluation to 50K generated samples, achieving an FID score of ***5.03323***. This result substantially outperforms LlamaGen [2], which reports FID scores of 7.15 and 6.09 after 100 and 300 epochs of training, respectively.
>
> We also conduct experiments on CelebA benchmark, which is single domain containing face images. We train the model  DiT-M/4 (layer=12, dim=512, num_head=8, and patch_size=4, similar to U-ViT [1]) using cosine and Laplace schedule. The training protocol consisted of 150,000 steps with a batch size of 256, employing minimal data augmentation (random flip only). The comparative FID results are as follows:
>
> | FID            | 100k    | 150k    |
> | -------------- | ------- | ------- |
> | cosine         | 10.0696 | 7.93795 |
> | Laplace (ours) | 7.93795 | 6.58359 |
>
> We also follow Stable Diffusion 3[1], train on a more comprehensive dataset CC12M[2] and report the FID results here. We download the dataset using `webdataset`. We train a DiT-base model using CLIP as text conditioner. The input images were preprocessed by cropping and resizing to $256\times 256$ resolution, then compressed into $32\times32\times4$ latent representations. The model training was conducted for 200K iterations using a batch size of 256.
>
> | FID            | 200k             |
> | -------------- | ---------------- |
> | Cosine         | 58.3619          |
> | Laplace (ours) | 54.3492(-4.0127) |
>
> The experimental results demonstrate that our proposed Laplace schedule achieves superior performance compared to  Cosine schedule. Specifically, our method yields an FID score of 54.3492, representing a substantial improvement of 4.0127 points over the baseline (58.3619).
>
>
> ***Q3: Visual results to demonstrate that this sampling method does not adversely affect the generated outputs.***
>
> FID quantifies the statistical similarity between model-generated samples and real-world reference data, where lower scores indicate more similarity to real data. It has been widely used to demonstrate the capability of visual generative models. The images sampled from our method are not adversely affected. We present some randomly generated cases in appendix (Figure 8) in modified version. In this figure, we also present results at different training iterations to demonstrate the fast convergence of our method.
>
> About the format of Table 1 and Table 2, we have modified the format in the revised PDF.
>
> [1] Bao, Fan, et al. "All are worth words: A vit backbone for diffusion models." CVPR. 2023.
>
> [2] Sun P, Jiang Y, Chen S, et al. Autoregressive Model Beats Diffusion: Llama for Scalable Image Generation[J]. arXiv preprint arXiv:2406.06525, 2024.

---

### Official Review · Reviewer_fzj6 · 2024-10-31

**Soundness:** 2
**Presentation:** 1
**Contribution:** 2
**Rating:** 3
**Confidence:** 3

**Summary:**

The paper analyzes the effect of various noise schedule choices for training diffusion models for ImageNet generation.  The authors show that noise schedule hyperparameter optimization favors putting more samples at times when the signal-to-noise ratio (SNR) is near one, and they found the model trained with Laplace schedule worked best in their experiments.  They evaluated the performance of the trained generative models by their FID score on ImageNet class-conditional image generation.

**Strengths:**

- Overall the paper's main contribution is an empirical investigation of a set of noise schedules.  The observation that training with more samples near SNR = 1 works better in their experiments could be relevant to practice.

**Weaknesses:**

- The results and claims are not significant contributions in my view, and some claims are unsupported.  In general, the paper lacks theoretical backing for their chosen noise schedule and the set of examined noise schedules.  The lack of theoretical grounding could be overcome by strong empirical analysis, but the empirical results show modest FID gains on ImageNet and are not conclusive enough to conclude this noise schedule should always be preferred.  If the primary argument is empirical, it would be good to see the claims generalizes across datasets.  Similarly, the claim that adjusting loss weights is worse than using a different noise schedule is too broad, as they did not optimize over different loss weighting schemes.

- The writing was also confusing.  For example, Equation 2 and Equation 7 use conflicting notation to describe the same loss optimization.  Also the title of section 2.2 includes the word "improved" but was background material on converting between time and SNR sampling; Section 2 did not clarify what was novel to the paper versus already known.  For instance, that the loss can be written in terms of sampling SNR and that loss weights are equivalent to different sampling is apparent and discussed in prior research.  The choice of figures is also surprising as figure 1 shows different distributions over log SNR but the paper contains no generated images displaying qualitative results.

**Questions:**

- Can the authors provide more details on the distinction between the contents of Table 1 and Table 2?

- The description of the inference strategy in Section 2.4 was brief but is important in understanding the results.  Can you expand on the precise procedure used at sampling time?

- The advantage of Laplace over Cosine Scaled in Table 3 appears limited.  Cosine Scaled is more stable with respect to CFG than Laplace, while almost meeting the optimized performance of Laplace.  Why suggest Laplace over Cosine Scaled given that Cosine Scaled seems more robust?

- In Figure 2, why are the other noise schedules not included?

---

> ### Author Response · Authors · 2024-11-21
>
> Thank you for your comments. We address your questions or concerns below.
>
> ***Q1: Distinction between Table 1 and Table 2.***
>
> Table 1 shows *the specific mathematical formulation of noise schedules*. $\lambda=\log \text{SNR}$ can be used to calculate the coefficients to add noise and $p(\lambda)$ is the distribution of $\lambda$.
>
> Table 2 show the choices of $p(\lambda)$ and $w(\lambda)$ *under the unified framework VDM++ by Kingma* et al [1]. For example, if we train diffusion models using cosine schedule and constant loss weight, the underlying $w(\lambda)$ is $e^{-\lambda / 2}$ and $p(\lambda)$ is $\text{sech} (\lambda / 2)$.
>
>
> ***Q2: Details about the sampling process.***
>
> For the detailed sampling process, *the sampled signal-to-noise ratios (SNRs) are aligned across different noise schedules*. Due to varying noise schedules, the same timestep $t$ corresponds to different SNRs under different noise schedules, as shown in Table 1. Specifically, for the cosine noise schedule, we utilize a uniformly sampled set of timesteps during the generation process: $\{t_0, t_1, \ldots, t_s\}$, which correspond to the SNRs: $\{\text{SNR}_0, \text{SNR}_1, \ldots, \text{SNR}_s\}$. For other noise schedules, such as the Laplace schedule, we compute the corresponding timesteps $\{t'_0, t'_1, \ldots, t'_s\}$ based on the SNRs: $\{\text{SNR}_0, \text{SNR}_1, \ldots, \text{SNR}_s\}$. This computation is feasible because the relationship is defined by a specific mathematical form and the function is monotonic. In practice, when using DDIM for sampling with the Laplace schedule, the timesteps employed are $\{t'_0, t'_1, \ldots, t'_s\}$.
>
>
> ***Q3: Why is the Laplace schedule recommended over the Cosine-Scaled schedule?***
>
> We adopt such schedule for several reasons. 1. The optimal performance is slightly better than cosine-scaled. 2. $p(\lambda)$ of Laplace is similar to that of cosine schedule with importance sampling as shown in Figure 4. 3. As we summarize in Sec 3.5, the specific mathematical formula is not that important. Even though the performances of these two schedules are superior to standard cosine schedule. , we want to offer a new choice to the community.
>
> ***Q4: Why not include other noise schedules in Figure 2?***
>
> In Figure 2, our primary objective is to demonstrate the *superior effectiveness of our noise schedule design compared to existing loss weight approaches*. Both Min-SNR and Soft-Min-SNR strategies assign higher weights to intermediate noise intensities and can serve as robust benchmarks. Including additional schedules would potentially obscure the key comparison and complicate the visual interpretation of the results.
>
> ***Q5: Limited performance improvement and Generalization to other datasets***
>
> In the context of class-conditional generation on ImageNet, improvement from 10.85 to 7.96 cannot be seemed as limited improvement. We extend our sample images to 50K to calculate the FID score. We obtain a score of 5.03. Notably, our base model, trained for merely 100 epochs, outperforms the advanced LLamaGen [2] model (which achieves FID scores of 6.09 and 7.15 at 300 and 100 epochs, respectively).
>
> To further validate the effectiveness of our proposed approach, we conduct additional experiments on the CelebA facial dataset, with results presented below:
>
> | FID $\downarrow$ | 100k    | 150k    |
> | ---------------- | ------- | ------- |
> | Cosine           | 10.0696 | 7.93795 |
> | Laplace (ours)   | 7.93795 | 6.58359 |
>
> Additionally, we extend our evaluation to the larger-scale CC12M dataset:
>
> | FID $\downarrow$ | 200k             |
> | ---------------- | ---------------- |
> | Cosine           | 58.3619          |
> | Laplace (ours)   | 54.3492 (-4.0127) |
>
> The experimental results from both unconditional image generation on CelebA and text-to-image generation on CC12M demonstrate the consistent superiority of our approach across different domains and tasks.

---

> > ### Author Response · Authors · 2024-11-21
> >
> > ***Q6: Choices of different loss weight schemes***
> >
> > The loss weight design, such as Min-SNR ($\gamma=5$), has demonstrated superior performance and we directly adopt the default setting from the paper.
> >
> > To further validate our findings, we conduct additional experiments the compare the noise schedule and derived loss weight from it. We choose Cosine-Ply ($n=2$, Equation 22), because it is directly based on cosine schedule.
> >
> > |         | Cosine (baseline) | Cosine-Ply (n=2) | Cosine-Ply as weight |
> > | ------- | ----------------- | ---------------- | -------------------- |
> > | FID-10K | 10.85             | 7.98             | 8.88                 |
> >
> > While the Cosine-Ply loss weight demonstrates improved performance over the standard cosine schedule, its FID score (8.88) remains inferior to the model trained with Cosine-Ply as a noise schedule (7.98).
> >
> >
> > ***Q7: Conflicts between equation 1 and equation 2.***
> >
> > These two equations are *not contradictory*. Different prediction targets, such as $\epsilon$ and $\mathbf{v}$, can be transformed into each other. $\boxed{\|\mathbf{v} - \hat{\mathbf{v}}\|_2^2 = (e^\lambda + 1)\|\mathbf{x} - \hat{\mathbf{x}}\|_2^2 = (e^{-\lambda} + 1)\|\boldsymbol{\epsilon} - \hat{\boldsymbol{\epsilon}}\|_2^2}$. Equation 2 presents a more unified framework. In practical model training, Equation 1 specifically uses $\mathbf{v}$ as the prediction target.
> >
> > ***Q8: Qualitative visual results comparison.***
> >
> > We add visual comparison results in Sec A.7 and Figure 8. Top rows show Cosine results, bottom rows show Laplace results, with five images per row at 100k-500k training iterations. Using identical noise inputs, our Laplace method shows faster convergence in both early object formation and detail refinement.
> >
> > ***Q9: About novelty.***
> >
> > Our innovative framework provides *a more unified perspective for analyzing noise schedules and importance sampling*. Based on this framework, we propose *a straightforward method for designing noise schedules*: one simply needs to identify a curve in the $p(\lambda)$ distribution and assign higher probabilities to intermediate levels.
> >
> > Our framework effectively explains existing approaches. Notably, our analysis aligns with the widely adopted flow matching with logit normal sampling method . Through our derivation (Sec A,4 and Figure 5), we discovered that flow matching with logit normal sampling assigns higher probabilities to intermediate steps compared to both cosine scheduling and vanilla flow matching. Recent state-of-the-art models, including Stable Diffusion 3 [3] and Movie Gen [4], have demonstrated enhanced training efficiency through the implementation of flow matching with logit normal sampling.
> >
> >
> > [1] Kingma D, Gao R. Understanding diffusion objectives as the elbo with simple data augmentation[J]. Advances in Neural Information Processing Systems, 2024, 36.
> >
> > [2] Sun P, Jiang Y, Chen S, et al. Autoregressive Model Beats Diffusion: Llama for Scalable Image Generation[J]. arXiv preprint arXiv:2406.06525, 2024.
> >
> > [3] Esser P, Kulal S, Blattmann A, et al. Scaling rectified flow transformers for high-resolution image synthesis[C]//Forty-first International Conference on Machine Learning. 2024.
> >
> > [4] Polyak A, Zohar A, Brown A, et al. Movie gen: A cast of media foundation models[J]. arXiv preprint arXiv:2410.13720, 2024.

---

> > > ### Comment · Reviewer_fzj6 · 2024-11-22
> > >
> > > Thanks for the reply.   I appreciate the effort to expand the experiments and empirical results which will improve the submission, solidifying claims around the observed usefulness of assigning higher probabilities near SNR=1.  However, I still feel the writing and contributions remain insufficient, so I have not changed my rating.

---

> > > > ### Author Response · Authors · 2024-11-26
> > > >
> > > > Dear Reviewer fzj6,
> > > >
> > > > We hope our above responses have addressed some of your concerns!
> > > >
> > > > We have made further updates to the writing as mentioned in our global response, improving the submission's flow, transitions, and contribution clarity.
> > > >
> > > > Regarding our contributions, we propose a unified framework that facilitates the design of noise schedules, leading to improved performance. Our approach also explains, to some extent, the effectiveness of other noise strategies, such as Flow Matching with logit normal sampling compared to the standard cosine schedule and vanilla flow matching.
> > > >
> > > > To our knowledge, this work presents a novel perspective, as no previous work has systematically explored this angle. The novelty, effectiveness, and ease to implementation are acknowledged by Reviewer **eM87**. If you are aware of similar findings that might challenge our claims, we would be more than willing to clarify the differences between our research and those findings.
> > > >
> > > > Here are our summarized contributions in revised introduction,
> > > >
> > > > >• A unified framework for analyzing and designing noise schedules in diffusion models, offering a more systematic approach to noise schedule optimization.
> > > >
> > > > >• Empirical evidence demonstrating the superiority of mid-range noise level focus over loss weight adjustments for improving training efficiency.
> > > >
> > > > >• Comprehensive evaluation and comparison of various noise schedules, providing practical guidelines for future research and applications in diffusion training.
> > > >
> > > >
> > > > We are dedicated to resolving your concerns. Thank you for your time and patience.
> > > >
> > > > Sincerely,
> > > > Authors of submission 3059

---

### Official Review · Reviewer_rSHk · 2024-11-04

**Soundness:** 2
**Presentation:** 1
**Contribution:** 2
**Rating:** 5
**Confidence:** 4

**Summary:**

This work aims to improve the efficiency of the training of diffusion models. The authors show that noise schedule is equivalent to importance sampling over noise intensities. Based on this, the authors derive some novel noise schedules to effectively assign more importance to mid-range noise intensities. The proposed method is experimentally demonstrated to be effective.

**Strengths:**

1.	The proposed method enjoys clean formulation.
2.	Judging from the experimental results provided, the proposed method shows consistent improvement over competitors in various scenarios.
3.	The formulation to relate noise schedule to noise importance sampling is rather universal, implying potential future extension.

**Weaknesses:**

Generally speaking, while the proposed method has the potential to make great contributions, the presentation of the manuscript makes it hard for the community to learn valuable new knowledge from this work. More concretely:

For Writing:

1. What is the most important key insight or takeaway for readers? While method details are extensive, there lacks a more general summary of the key insight, especially in the introduction section.
2. The authors should provide more background for noise schedule in the introduction section, letting readers know the actual meaning and context of noise schedule as early as possible.
3. What is the relation between the proposed method and the two previous approaches (loss weighting and architectural changes)? What inspires the authors to devise the new method given the existence of previous approaches?

For Evaluation:

4. The comparison of loss weighting and noise schedule might not be entirely fair. I would recommend converting the same distribution into both loss weighting and noise schedule for a more insightful comparison.
5. There can be some qualitative visualization evaluation. For example, showcasing image samples across different training steps with different methods.

**Questions:**

See weaknesses.

---

> ### Author Response · Authors · 2024-11-21
>
> Thank you for your feedback. We address your questions or concerns below.
>
> ***Q1:  Key insight / takeaway for readers.***
>
> **Takeaway**: This framework *simplifies noise schedule design* in diffusion models - instead of complex calculations, researchers can now achieve better results by simply identifying and emphasizing intermediate noise levels in the $p(\lambda)$ distribution.
>
> Our insights align with recent methods that employ flow matching with logit normal sampling. Specifically, we demonstrate how flow matching with logit normal sampling assigns higher intermediate step probabilities than both cosine scheduling and standard flow matching approaches. This aligns with empirical results from our analysis (detailed in Section A.4 and Figure 5). The effectiveness of this approach is evident in recent leading models like Stable Diffusion 3 [1] and Movie Gen [2], where it has improved training efficiency.
>
> ***Q2: Add more background about noise schedule.***
>
> >Noise schedule in diffusion models is a function that determines how much noise is added to the input data at each timestep $t$ during the training process, controlling the distribution of noise levels that the neural network learns to remove.
>
> We have added it in introduction in revised manuscript, highlighted in blue for easy reference.
>
> ***Q3: Relationship between architecture design and loss weight schemes and motivation to design new method.***
>
> Our method is *orthogonal to architecture design*. Our method shares similar insights like loss weight design but achieve superior performance. Specifically, using $\mathbf{v}$ as predict target, the Min-SNR / Soft-Min-SNR loss weight is $\frac{\min\{\text{SNR}, \gamma\}}{\text{SNR} + 1}$ / $\frac{1}{\text{SNR} + 1}\frac{1}{\text{SNR}^{-1} + \gamma^{-1}}$  respectively, which puts more weight on middle level noise densities (as shown in Figure 7).
>
> The loss weight determines the relative importance of each noise level in the training dynamics of diffusion models. Our approach directly allocates computational resources by assigning more training FLOPs to specific noise densities.
>
> We investigated the significance of training intervals in diffusion models by partitioning the time domain $(0,1)$ into four equal segments $\text{bin}_i = \left(\frac{i}{4}, \frac{i + 1}{4}\right), i=0,1,2,3$. After training a base model $\mathbf{M}$ across the entire range, we fine-tuned it on individual segments to create specialized checkpoints $\mathbf{m}_0$ through $\mathbf{m}_3$. The evaluation combined these specialized checkpoints within their corresponding time intervals while utilizing the base model elsewhere. FID measurements demonstrated that optimizing intermediate noise levels (segments 1 and 2) yielded superior performance, indicating that allocating more computational resources to mid-level noise densities enhances the diffusion model's effectiveness under fixed training constraints. The related results can be found in revised A.5.
>
> ***Q4: Comparison between noise schedule and loss weight from $p(\lambda)$***
>
> We have added experiments here to compare with noise schedule and it performed as loss weight. We choose Cosine-Ply as in Equation 22 because it is based on standard cosine schedule and can achieve relative low FID.
>
> |         | Cosine (baseline) | Cosine-Ply (n=2) | Cosine-Ply as weight |
> | ------- | ----------------- | ---------------- | -------------------- |
> | FID-10K | 10.85             | 7.98             | 8.88                 |
>
> We can see that even though such loss weight performs better than standard cosine schedule, but it still lags behind than that is trained as noise schedule.
>
> ***Q5: Visualization Results***
>
> We have provided several visual examples in revised manuscript. We compare the samples generated from DiT-B trained with Cosine schedule and Laplace schedule at different training iterations. Our results demonstrate faster convergence and better visual qualities which aligns with the FID results.
>
> [1] Esser P, Kulal S, Blattmann A, et al. Scaling rectified flow transformers for high-resolution image synthesis[C]//Forty-first International Conference on Machine Learning. 2024.
>
> [2] Polyak A, Zohar A, Brown A, et al. Movie gen: A cast of media foundation models[J]. arXiv preprint arXiv:2410.13720, 2024.

---

> > ### Comment · Reviewer_rSHk · 2024-11-22
> >
> > Thanks for the reply. These new experimental results have further consolidated the evaluation, and I believe the manuscript refinement will help the paper better convey important information.
> > I will raise the rating, acknowledging the authors' efforts. However, considering the state of the initial submission, this work might require more time for improvement. Hence, I still lean towards rejection.

---

> > > ### Author Response · Authors · 2024-11-26
> > >
> > > Dear Reviewer rSHk,
> > >
> > > We would like to express our deepest gratitude again for your invaluable feedback and the outstanding efforts you have invested in reviewing our work. In response to your comments, we have made further updates as outlined in our global response and highlighted the revised parts in blue in the latest PDF.
> > >
> > > Sincerely,
> > > Authors of submission 3059

---

### Official Review · Reviewer_eM87 · 2024-11-04

**Soundness:** 2
**Presentation:** 2
**Contribution:** 2
**Rating:** 3
**Confidence:** 3

**Summary:**

This paper presents a novel method for enhancing the training of diffusion models by strategically redefining the noise schedule. The key insight is the importance sampling of the logarithm of the Signal-to-Noise ratio (log SNR) theoretically equivalent to a modified noise schedule. Empirical results show that the proposed Laplace noise schedule, which focuses computational resources on mid-range noise levels, yields superior performance compared to adjusting loss weights under constrained computational budgets.

**Strengths:**

1. The paper presents a novel approach to design the noise schedule for enhancing the training of diffusion models.
2. The proposed method is effective and easy to apply.
3. The findings contribute to the ongoing efforts to optimize diffusion models, potentially paving the way for more efficient and effective training paradigms.
4. The paper is overall well-written and easy to follow.

**Weaknesses:**

1. The paper lacks a formal theoretical guarantee for the effectiveness of the proposed method. While the experimental results are promising, providing a theoretical foundation would strengthen the validity of the approach. It would be better to add more analysis on why Laplace schedule is working and why we should increase the sample frequency around $log SNR = 0$.
2. The experiments are only conducted on the ImageNet dataset with different resolutions. However, the generalizability of the proposed method to other datasets or tasks remains untested.
3. The effectiveness of this method seems not stable enough. According to Table 3, the proposed Laplace is ineffective when CFG=1.5.

**Questions:**

Please see the weakness section.

---

> ### Author Response · Authors · 2024-11-21
>
> Thanks for your feedback on our manuscript.  We address your questions or concerns below.
>
> ***Q1: Why does Laplace schedule work and why should we increase the sampling frequency.***
>
> We explored the importance of different training time intervals in a diffusion model by dividing the time range $(0,1)$ into four equal segments $\text{bin}_i = \left(\frac{i}{4}, \frac{i + 1}{4}\right), i=0,1,2,3$. A base model $\mathbf{M}$ was trained over the full range, followed by specialized fine-tuning on each segment to create four checkpoints $\mathbf{m}_0$ through $\mathbf{m}_3$. The evaluation process involved using these specialized checkpoints within their respective time intervals while defaulting to the base model elsewhere. Results from FID measurements across these configurations revealed that optimizing the middle level noise (segments 1 and 2) led to better performance, highlighting that under the same training budget, putting more compute to middle level noise densities contributes to better performance of the diffusion model. The related results can be found in revised A.5.
>
> Furthermore, we present an additional derivation for Flow Matching with logit-normal sampling. Research on scaling laws for diffusion transformers [4] reveals that the cosine schedule outperforms vanilla flow matching, while flow matching with logit-normal sampling demonstrates superior performance compared to the cosine schedule, which aligns with our discovering. From the logSNR distribution perspective, flow matching with logit-normal sampling exhibits greater concentration around logSNR=0 compared to both the cosine schedule and vanilla flow matching. We include the derivation and visualization of $p(\lambda)$ in revised Appendix A.4.
>
>
> ***Q2: Experiments on additional datasets.***
>
> ImageNet, comprising over one million natural images, has been widely used to validate improvements in diffusion models [2, 3]. To validate the generality of our method, we conduct additional experiments on two more datasets, including CelebA (single domain of face images) and CC12M (text to image dataset).
>
> Using the CelebA dataset at 64×64 resolution, we trained a DiT architecture in pixel space. The architecture comprises 12 layers with an embedding dimension of 512, 8 attention heads, and a patch size of 4, implemented across different noise schedules. We evaluated this unconditional single-domain generation model using the FID metric, with results shown below:
>
> | FID $\downarrow$ | 100k    | 150k    |
> | ---------------- | ------- | ------- |
> | cosine           | 10.0696 | 7.93795 |
> | Laplace (ours)   | 7.93795 | 6.58359 |
>
> Following stable diffusion 3[1], we train diffusion models on a more complicated dataset CC12M[2] dataset , which comprises over 12 million image-text pairs. We download the dataset using `webdataset`. We train a DiT-base model using CLIP as text conditioner. The images are cropped and resized to $256\times 256$ resolution, compressed to $32\times 32\times 4$ latents and trained for 200k iterations at batch size 256. The experimental results are presented below:
>
> | FID $\downarrow$ | 200k             |
> | ---------------- | ---------------- |
> | cosine           | 58.3619          |
> | Laplace (ours)   | 54.3492(-4.0127) |
>
> Our method's generalizability is demonstrated through its performance on both unconditional image generation (CelebA) and text-to-image generation (CC12M) tasks.
>
> ***Q3: Stability of results across different classifier-free guidance (CFG) scales.***
>
> In the context of classifier-free guidance, *a strong model does not necessarily achieve the lowest FID across different CFG scales.* Recent prominent diffusion models search the classifier-free guidance scale to demonstrate their best performance. For example, DiT [2] chooses CFG = 1.5 to report its best performance, while EDM2 [3] selects CFG = 1.4. We conducted sampling at three CFG values and selected the optimal one to *better evaluate the actual performance of different models*. The model trained with cosine schedule favors CFG=2.0 while the model trained with Laplace schedule favors CFG=3.0.
>
> [1] Bao, Fan, et al. "All are worth words: A vit backbone for diffusion models." CVPR. 2023.
>
> [2] Peebles W, Xie S. Scalable diffusion models with transformers[C]//Proceedings of the IEEE/CVF International Conference on Computer Vision. 2023: 4195-4205.
>
> [3] Karras T, Aittala M, Lehtinen J, et al. Analyzing and improving the training dynamics of diffusion models[C]//Proceedings of the IEEE/CVF Conference on Computer Vision and Pattern Recognition. 2024: 24174-24184.
>
> [4] Liang Z, He H, Yang C, et al. Scaling Laws For Diffusion Transformers[J]. arXiv preprint arXiv:2410.08184, 2024.

---

> > ### Comment · Reviewer_eM87 · 2024-11-27
> >
> > Thanks for the authors' response, which addressed some of my initial concerns. However, after considering the feedback from the other reviewers and revisiting the paper, I agree with some of their points, particularly regarding the need for improvements in the writing and the fact that the technical contributions remain insufficient. Taking all this into account, I have decided to lean toward rejection.

---

> ### Author Response · Authors · 2024-11-27
>
> Thanks for your reply.
>
> We have conducted additional experiments and substantially revised the manuscript to address your feedback. While we note some discrepancies between your initial and current assessment (with lower rating), we would like to reference your earlier evaluation that highlighted several strengths of our work:
>
> > The paper presents a ***novel approach*** to design the noise schedule for enhancing the training of diffusion models.
> > The proposed method is ***effective and easy to apply***.
> > The findings ***contribute to the ongoing efforts to optimize diffusion models***, potentially paving the way for more efficient and effective training paradigms.
> > The paper is ***overall well-written and easy to follow***.
>
> We have carefully addressed your current concerns in the *revised manuscript* and look forward to your evaluation of these improvements. We remain committed to resolving any remaining issues within the revision timeline.
>
> Best,
> Authors of submission 3059

---

### Official Review · Reviewer_i2Dm · 2024-11-05

**Soundness:** 2
**Presentation:** 2
**Contribution:** 2
**Rating:** 5
**Confidence:** 2

**Summary:**

This paper presents an approach to improving the training efficiency of diffusion models, focusing on the design of the noise schedule. The authors propose a strategy based on importance sampling the logarithm of the Signal-to-Noise Ratio (log SNR), specifically concentrating on noise levels around log SNR = 0. This targeted focus on the transition between signal dominance and noise dominance helps improve the model’s ability to make more robust predictions.

The paper compares various noise schedules (Laplace, Cauchy, Cosine Shifted, and Cosine Scaled) and empirically shows that the Laplace noise schedule outperforms the traditional cosine schedule, achieving better results in terms of training efficiency. The proposed method demonstrates its effectiveness on benchmarks like ImageNet and provides insights into optimizing diffusion model training under constrained computational resources.

**Strengths:**

1. The paper proposes a noise schedule design, focusing computational resources on medium noise levels (log SNR = 0). It provides a perspective on optimizing diffusion model training, potentially paving the way for more efficient generative AI methods.

2. The authors conduct a thorough set of experiments, comparing various noise schedules and loss weighting strategies. The experimental setup is robust, testing the methods on multiple resolutions and prediction targets, and demonstrating Laplace schedule’s performance.

3. By improving training efficiency and robustness under constrained computational budgets, this work addresses a challenge for applying diffusion models to large-scale applications like high-resolution image synthesis and long video generation.

**Weaknesses:**

1. While the experiments focus on ImageNet and high-resolution image tasks, the paper does not explore how the proposed noise schedules would perform in other domains or more complex real-world scenarios which would be useful for understanding the broader applicability of the approach.

2. Figure/tables might not be fully convincing, and it would be great to explain if they are statistically significant. In Figure 2, we could see that as number of training iterations increase to 500k, the gap between the proposed approach and benchmark shrinks significantly, so a nature question is if this still remains superior as number of training iterations increase more, beyond 500k. In Table 3, it shows that Laplace has the best performance when CFG=3.0, but that is not the case for other CFG values. Should we expect them to outperform for any given CFG value or should we anchor only on its performance at CFG=3.0. If only at CFG=3., how does it impact its adaptivability to different settings or datasets?

**Questions:**

While the results on ImageNet are promising, further experiments on other types of data could help validate the generalizability of the approach. also, what would the impact of the proposed noise schedule be on diffusion models trained in more dynamic or non-stationary environments? It would be useful to explore how the noise schedule adapts in settings where the distribution of data might shift.

---

> ### Author Response · Authors · 2024-11-21
>
> Thank you for your thoughtful feedback. We address your questions or concerns below.
>
> ***Q1: Performance in other domains or more complex real-world scenarios***
>
> ImageNet, comprising over one million natural images, has been widely adopted as a benchmark dataset for validating improvements in diffusion models [4, 5, 6].
> In addition to ImageNet, we evaluate our approach on the CelebA dataset (64×64 resolution in pixel space), which consists of face images. We employ a DiT architecture (12 layers, embedding dimension of 512, 8 attention heads, and patch size of 4) using different noise schedules. This is an unconditional generation setting within a single domain. We present FID results as follows:
>
> | FID $\downarrow$ | 100k    | 150k    |
> | ---------------- | ------- | ------- |
> | cosine           | 10.0696 | 7.93795 |
> | Laplace (ours)   | 7.93795 | 6.58359 |
>
> We also follow Stable Diffusion 3 [1], train on a more complicated dataset CC12M[2] dataset (over 12M image-text pairs) and report the FID results here. We download the dataset using `webdataset`. We train a DiT-base model using CLIP as text conditioner. The images are cropped and resized to $256\times 256$ resolution, compressed to $32\times 32\times 4$ latents and trained for 200k iterations at batch size 256.
>
> | FID $\downarrow$ | 200k             |
> | ---------------- | ---------------- |
> | cosine           | 58.3619          |
> | Laplace (ours)   | 54.3492(-4.0127) |
>
> Our method demonstrated strong generalization capabilities across both unconditional image generation using the CelebA dataset and text-to-image generation using the CC12M dataset.
>
> ***Q2: Gap after longer training.***
>
> As we presented in the paper, our goal is to *obtain a strong diffusion model under limited and fixed training budget*. With longer training, the gap is going to be smaller.  That is also because of our model *already achieves relative good results*. We generate more samples and extend FID-10k to FID-50k, the score comes to  ***5.03323.*** Even with only 100 epoch training, our model outperforms LlamaGen [3] with longer training (FID 7.151 at 100 epoch, FID 6.092 at 300 epoch, in table 9 [3]).
>
> ***Q3: Effects of CFG selection and performance variability across different CFG values***
>
> A strong model does not mean the best FID scores over different CFG values. Recent famous diffusion models search the classifier-free guidance scale to demonstrate their performance, *e.g.*, DiT [6] choose 1.5 to report their performance and EDM2 [7] chooses 1.4. We conduct sampling on three cfg values to and choose the best one to *better demonstrate the performance of different models*. The model trained with cosine schedule favors CFG=2.0 while the model trained with Laplace schedule favors CFG=3.0. For different models, different training datasets, the suitable classifier free guidance differs and we should try different values instead of directly using a pre-defined value.
>
> [1] Esser P, Kulal S, Blattmann A, et al. Scaling rectified flow transformers for high-resolution image synthesis[C]//Forty-first International Conference on Machine Learning. 2024.
>
> [2] Changpinyo S, Sharma P, Ding N, et al. Conceptual 12m: Pushing web-scale image-text pre-training to recognize long-tail visual concepts[C]//Proceedings of the IEEE/CVF conference on computer vision and pattern recognition. 2021: 3558-3568.
>
> [3] Sun P, Jiang Y, Chen S, et al. Autoregressive Model Beats Diffusion: Llama for Scalable Image Generation[J]. arXiv preprint arXiv:2406.06525, 2024.
>
> [4] Karras T, Aittala M, Aila T, et al. Elucidating the design space of diffusion-based generative models[J]. Advances in neural information processing systems, 2022, 35: 26565-26577.
>
> [5] Dhariwal P, Nichol A. Diffusion models beat gans on image synthesis[J]. Advances in neural information processing systems, 2021, 34: 8780-8794.
>
> [6] Peebles W, Xie S. Scalable diffusion models with transformers[C]//Proceedings of the IEEE/CVF International Conference on Computer Vision. 2023: 4195-4205.
>
> [7] Karras T, Aittala M, Lehtinen J, et al. Analyzing and improving the training dynamics of diffusion models[C]//Proceedings of the IEEE/CVF Conference on Computer Vision and Pattern Recognition. 2024: 24174-24184.

---

### Author Response · Authors · 2024-11-26
**Global Response (1/2)**

Dear AC and reviewers,

We sincerely thank the reviewers for their valuable and constructive feedback to help improve our submission! We are encouraged by the recognition of our work's contributions: Multiple reviewers acknowledged the importance of addressing training efficiency in diffusion models and appreciated our thorough experimental validation. Reviewer **eM87** highlighted the ease of implementation, while Reviewer **rSHk** noted our clean formulation. Reviewer **i2Dm** commended our approach's effectiveness under computational constraints. Reviewers **rSHk** and **fzj6** recognized the value of our unified framework and empirical insights. Reviewers **i2Dm**, **eM87**, and **f1Vg** appreciated our method's consistent performance improvements across various experiments.

We have addressed each reviewer's questions individually. Below, we present a concise summary of our responses to some common concerns:

1. **Additional Experiments on Other Datasets**. We conducted additional experiments on CelebA (unconditional generation) and CC12M (text-to-image generation), demonstrating consistent improvements across different domains and tasks. These experiments further demonstrate the *generalization of our method*.
2. **Visual Results and Evaluation**. We showcase our method's *fast convergence* by displaying results at various training iterations using randomly generated cases in the appendix. We also extended our quantitative evaluation to 50K samples (achieving FID 5.03), outperforming recent methods like LlamaGen even with fewer training epochs.
3. **Stability and CFG Analysis**. We clarified that different models naturally favor different CFG values, as acknowledged in recent literature. We experimented with different CFG scales to better illustrate the model's performance.
4. **Comparison with Loss Weighting**. We conducted additional experiments comparing noise schedules with their equivalent loss weight implementations. Results show that directly modifying the noise schedule (FID 7.98) is more effective than implementing it as a loss weight (FID 8.88).
5. **Writing Improvements**. We revised the introduction to better explain the role of the noise schedule in diffusion models and to strengthen the claim of our contribution. We also improved the method section by adjusting titles for better suitability and ensuring smoother transitions between sections.

We have carefully considered each reviewer's feedback and made corresponding revisions. Here is a summary of updates (marked blue in revised PDF):

In the main paper,

1. In the introduction, we revised the second and third paragraphs to simplify the discussion on structural design and loss weight design. This revision ensures a more concise presentation without delving too deeply into related work, resulting in a smoother narrative flow.
2. In the introduction, we provided a *clearer summary of our contributions* compared to the previous version. This revision makes it easier for readers to follow and understand our key points.
3. In the method section 2.1, we added an explanation for why we study noise schedules in the PRELIMINARIES section.
4. In the method section 2.2, we changed the subtitle from `Improved Noise Schedule Design` to `Noise Schedule Design from a Probability Perspective.` This makes the section more specific and adds an explanation of the generality of our approach, providing a more intuitive understanding of existing noise schedules.
5. In the method section 2.3, we explained the purpose of introducing a unified framework: within this framework, we can now apply these theoretical insights to practical settings.
6. In the method section 2.4, we added a sentence to make our sampling process description more specific.
7. In section 3.1, we revised the text to explain why we tested multiple CFG values and their corresponding FID scores.
8. In Section 3.5, we further summarized the experimental results: While these different schedules take various mathematical forms, they all achieve similar optimal performance when given equivalent training budgets. The specific mathematical formulation is less crucial than the underlying design philosophy: increasing the sampling probability of intermediate noise levels. This principle provides a simple yet effective guideline for designing noise schedules.
9. For Table 1, we changed the format to align with Table 2.

---

> ### Author Response · Authors · 2024-11-26
> **Global Response (2/2)**
>
> In the appendix,
>
> 1. In A.4, we added a derivation of the distribution $p(\lambda)$ for Flow Matching with Logit Normal Sampling, a recently popular training sampling method, and compared it with the cosine schedule and vanilla flow matching. We explained its superiority from the perspective of $p(\lambda)$.
> 2. In A.5, we conducted experiments to verify which intervals are more important for diffusion learning. We confirmed that learning intermediate noise strengths is more crucial than learning at the extremes.
> 3. In A.6, we added experiments comparing noise schedules with implementing them as loss weights, demonstrating the effectiveness of noise schedules.
> 4. In A.7, we added experiments on more datasets, including unconditional generation on the CelebA dataset, which contains only facial data, and conditional generation on the more complex text-image dataset CC12M. Our approach was effective in both cases, demonstrating its generalizability.
> 5. In A.8, we added more visual comparisons, showing that our approach not only performs well in FID metrics but also presents superior results and convergence speed in generated samples.
>
> Best,
> Authors of submission 3059

---

### Meta-Review · Area_Chair_uhpw · 2024-12-17

**Metareview:**

This paper studies noise schedule for training diffusion models that improves efficiency by increasing the sampling frequency around a log SNR of 0, leading to faster convergence and enhanced performance. After the rebuttal phase, the overall comments from our reviewers are negative. Some concerns still need to be addressed, and two major ones are the need for improved writing and the weakness of technical contributions. I have to recommend rejecting this paper, hoping reviewers' feedback will be helpful for future submissions.

**Additional Comments On Reviewer Discussion:**

`i2Dm`: borderline (5). I didn't take part in the rebuttal, which focuses on the significance of the experiments on more complicated scenarios and the performance gap for longer training. The advantage over the beginning phase is sufficient to show efficacy since faster convergence often corresponds to improved efficiency. However, I agree with the reviewer that it should be tested on more datasets.

`eM87`: negative (3) after rebuttal, who thinks it needs writing improvements and the technical contributions remain insufficient.

`rSHk`: borderline (5) after rebuttal,  who thinks this work might require more time for improvement considering the state of the initial submission

`fzj6`: negative(3), who thinks the writing is confusing and experiments are not significant

`f1Vg`: borderline (5), who thinks there is a lack of theoretical justification and the experiments are insufficient.

Almost all reviewers pose the similar major concerns: 1) technical and theoretical contribution 2) writing 3) experimental significance. I have to recommend rejecting it.

---

### Decision · Program_Chairs · 2025-01-22

Reject